# Improving Generative Adversarial Networks via Adversarial Learning in Latent Space

**Yang Li[1], Yichuan Mo[12], Liangliang Shi[1], Junchi Yan[1]\***
[1]Department of Computer Science and Engineering,
MoE Key Lab of Artificial Intelligence, Shanghai Jiao Tong University
[2]Key Lab. of Machine Perception (MoE),
School of Intelligence Science and Technology, Peking University
`{yanglily,shiliangliang,yanjunchi}@sjtu.edu.cn`
`mo666666@stu.pku.edu.cn`

## Abstract

For Generative Adversarial Networks which map a latent distribution to the target distribution, in this paper, we study how the sampling in latent space can affect the generation performance, especially for images. We observe that, as the neural generator is a continuous function, two close samples in latent space would be mapped into two nearby images, while their quality can differ much as the quality generally does not exhibit a continuous nature in pixel space. From such a continuous mapping function perspective, it is also possible that two distant latent samples can be mapped into two close images (if not exactly the same). In particular, if the latent samples are mapped in aggregation into a single mode, mode collapse occurs. Accordingly, we propose adding an implicit latent transform before the mapping function to improve latent $z$ from its initial distribution, e.g., Gaussian. This is achieved using well-developed adversarial sample mining techniques, e.g. iterative fast gradient sign method (I-FGSM). We further propose new GAN training pipelines to obtain better generative mappings w.r.t quality and diversity by introducing targeted latent transforms into the bi-level optimization of GAN. Experimental results on visual data show that our method can effectively achieve improvement in both quality and diversity. The implementation is publicly available at `https://github.com/yangco-le/AdvLatGAN`.

## 1 Introduction

Generative Adversarial Networks (GANs) [1] have shown effectiveness for generating high-fidelity data, especially for images under various settings [2; 3; 4; 5; 6]. Based on the zero-sum game, the model is trained by the adversarial process between the generator and the discriminator. Many efforts have been made to achieve a more realistic generation from different perspectives. For instance, WGAN [7], SNGAN [8], LSGAN [9] aim to design better objective functions. ImprovedGAN [10], AC-GAN [11] propose practical training techniques. While more complex network structures are studied in BigGAN [12], StyleGAN [13], SAGAN [14].

Despite the above progress, a relatively less studied question is how the latent space affects the generation quality and diversity, which can be orthogonal to the above frequently studied factors like model architectures and training techniques. In this paper, we argue that the latent samples from a standard continuous distribution (e.g. Gaussian) can often be mapped to varying-quality samples for the generation. One reason is that the generator is a continuous function (as neural nets)

---

\*Junchi Yan is the correspondence author who is also with Shanghai AI Laboratory.

36th Conference on Neural Information Processing Systems (NeurIPS 2022).

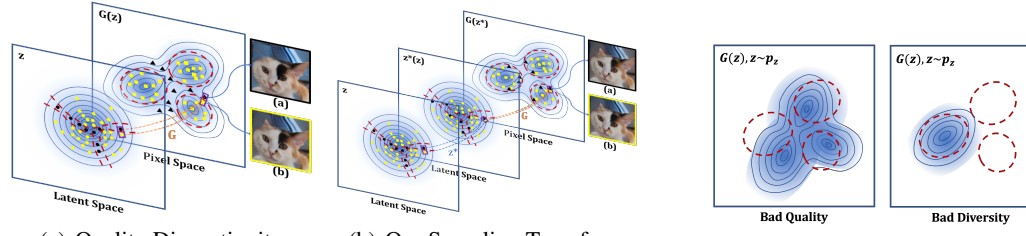

| (a) Quality Discontinuity | (b) Our Sampling Transform |
|---|---|

Figure 1: Red dashed lines: real data manifolds and their preimages; yellow circles/squares: samples for sound generation; black circles/squares: samples for bad generation; orange dashed lines: the generator mapping; blue dashed lines: $z$ transform; blue shades and solid blue lines: latent distribution and generated distribution. Due to the continuity of the network-based generator, close samples in latent space will be mapped to close images in pixel space, while the quality can vary. Cat (a) and Cat (b) are close in pixel space, but Cat (a) loses its right ear. Our method transforms bad latent $\mathbf{z}$ towards the preimage of the nearest real manifold, thus avoiding terrible generation.

Figure 2: The generated distributions mapped from a fixed latent distribution reflect the mapping quality (same legends as in Fig. 1). Left: bad generation quality as the generated distribution does not match the real distribution well. Right: bad generation diversity as the mapping misses modes.

while the generation quality reflected by the matching degree to the natural image distribution in pixel space does not exhibit a continuous nature as two nearby images' quality can differ much. It has been shown in GAN literature [15] that image objects lie upon multiple disjoint manifolds. A continuous mapping from a continuous latent distribution, e.g., Gaussian, may generate invalid samples in the between of the manifolds when the generator seeks to cover all the meaningful modes. See Fig. 1 (a) for illustration. This fact also implies a dilemma: for complex real-world data, there is a trade-off between quality and diversity when the latent vectors are sampled from a continuous uni-mode distribution. On the other hand, to effectively capture the natural image distribution through a continuous generative mapping, it is necessary to maintain a disconnected latent space support.

The above contradiction between the discontinuity in generation data quality and the continuity of latent distribution as well as mapping function, can be one of the significant factors leading to the well-known training difficulty/instability of GANs, especially for generators, as one unreasonably hopes to enforce a discontinuous mapping via continuous neural nets. On the other hand, the mapping is also critical to the performance of functions with continuous nature, as badly trained mapping can lead to poorly fitted generated distribution. Meanwhile, by realizing that the mapping function allows for many-to-one mapping, mode collapse [16] can naturally happen when different latent samples are mapped into few or even a single mode in the target space. See Fig. 2 for the schematic diagram.

To tackle these issues, we impose an extra (implicit) transform function $z^*(\cdot)$ on the raw sampling $\mathbf{z}$, and then the generation can be written by $G(z^*(\mathbf{z}))$ which requires us to make **twofold efforts on both $G(\cdot)$ and $z^*(\cdot)$**. From this perspective, most existing works employ $z^*(\cdot)$ as an identity function, while we try to find a more effective one. We model the $\mathbf{z}$ transform process as making perturbations to the original sampling since $z^*(\mathbf{z})$ shall not depart much from $\mathbf{z}$ as we hope the main content of the generated image remains the same. The hope that small perturbations can achieve considerable positive quality variation leads us to the adversarial sample mining methods. Specifically, the implicit function based transform $z^*(\cdot)$ can be achieved by updating the raw latent distribution samples $\mathbf{z}$ to minimize $loss_G$ i.e. $\log(1 - D(G(\mathbf{z})))$ using adversarial sample mining techniques e.g. iterative fast gradient sign method (I-FGSM) [17], and we name the method as AdvLatGAN-z. See Fig. 1 (b) for the schematic diagram. We will show that our $\mathbf{z}$ transform (perturbations) can deliver a latent distribution better fitting the real distribution with the fixed $G$.

On top of our first effort, we propose a training strategy to improve the generative mapping $G$ respectively for quality and diversity. This is achieved by introducing $\mathbf{z}$ optimization (i.e., targeted implicit transforms on $\mathbf{z}$) into GAN training, specifically, updating raw latent distribution samples $\mathbf{z}$ using I-FGSM to find latent variables which benefit the optimization to calculate the loss during GAN training. Respectively for quality and diversity, we use two iterative updating strategies for

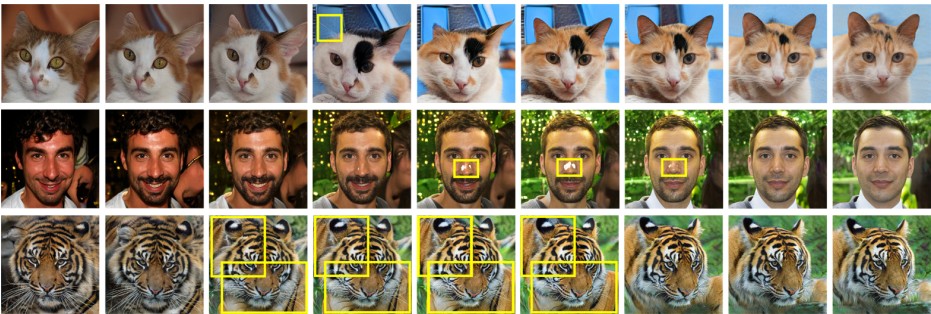

Figure 3: Generation examples of StyleGAN2-ada [18]. Sampling equidistantly from $p(\mathbf{z})$ in latent space, the samples from the generated distribution in pixel space are shown here. The beginning and ending images of each row locate in different manifolds. The intermediate results are sampled in the between of disjoint manifolds. The discontinuity causes weird generation (see yellow boxes – **top**: missed ear and crooked head; **middle**: white spots on the nose; **bottom**: two bodies share one head).

**z** under different objectives, and we name the two training algorithms as AdvLatGAN-qua and AdvLatGAN-div. **The highlights of this paper include:**

1) We rethink the role of the generator as a continuous function, which may incur quality discontinuity in target space when the latent samples are directly sampled from a continuous distribution, together with poorly fitted generated distribution and mode aggregation (collapse) when the function suffers bad properties w.r.t. quality or diversity. We propose to formulate the mapping from $G(\mathbf{z})$ to $G(z^*(\mathbf{z}))$. It provides new design guidance to GAN which requires efforts on both $G(\cdot)$ and $z^*(\cdot)$.

2) By introducing adversarial techniques into GAN, we propose a sampling transform pipeline for $z^*(\cdot)$ and training pipelines for improving $G(\cdot)$. The generation results are obtained by first training generative mappings by our proposed training pipeline AdvLatGAN-qua/div. Then the transform $z^*(\cdot)$ namely AdvLatGAN-z is enforced and computed implicitly to obtain an improved distribution for a more realistic generation. The training algorithms AdvLatGAN-qua and -div involve latent sample transform under different objective functions for quality and diversity.

3) Experimental results on synthetic data and natural images including large-scale datasets e.g. ImageNet show the notable improvements of our approach on both generation quality and diversity, whereby the performance gain is basically orthogonal to those from existing GAN techniques.

## 2 Related Work

Our latent vector transform pipeline mines effective latent samples and the resulting perturbations relate to adversarial methods. The proposed GAN training pipelines include the aforementioned transform in network training, which can be viewed as mining adversarial samples in training, namely adversarial training. Hence we briefly review existing works on latent space mining for GANs that modify latent space samples to achieve specific targets e.g. latent space sampling improvement for better generation and adversarial learning. Also, techniques for avoiding mode collapse are studied.

**Latent space mining in GANs.** Techniques have been devised for latent space mining for GANs. By utilizing the steerability of GANs in latent space, StyleGAN [13] introduces a latent variable shift to better decouple the various styles by using a linear network (which is still continuous). [19], [20] and [21] introduce the latent space mining to training to achieve clustering objectives or improve the dynamic of training. [22] and [23] achieve conditional generation by performing gradient ascent in latent space to maximize classification neuron activations. Some recent works focus on improving latent sampling quality to obtain high-quality images. Interactive evolutionary computation is used to make the generation process more controllable in [24], which meanwhile also brings additional manual efforts. To reduce manual intervention, [25; 26] use Koncept512 [27] as a criterion to improve the generation quality. However, as discussed in [25], this strategy is somehow biased and can not work well for each dataset (e.g., Pokemons). The reason is that the quality estimator is not even designated to access the real-data distribution information. Instead, the generation is guided by a predefined score function. Exploring the real distribution's guidance from the discriminator, DOT [28] utilizes optimal transport in latent space to improve the initial sampled

distribution, DRS [29] proposes a rejection sampling scheme to to improve sampling quality, and DDLS [30] develops discriminator driven Langevin dynamics to obtain high-quality samples.

**Mitigating mode collapse for GANs.** Efforts are made to mitigate mode collapse, which are based on the thinking that real-world samples lie in disjoint manifolds [15]. One way is to introduce a collection of generators such that each generator may cover a specific mode since a single generator can hardly fulfill a discontinuous function [15]. However, more works focus on improving mapping properties, explicitly obtaining a mapping covering more modes [31; 32; 33]. Note that the generated distribution obtained by this method will contain invalid parts due to the continuity of the latent distribution and the mapping. This calls for the selective sampling of the continuous latent distribution or directly amending the latent distribution, which is rarely noticed in these works.

**Adversarial samples and adversarial training.** Adversarial samples are manipulated samples crafted by adding indistinguishable perturbations to cause significant deep nets' output variation e.g. wrong classification prediction. Adversarial sample mining methods investigate how to manipulate such samples, which are known as adversarial attacks. Since [34], various adversarial attack methods have been devised [35; 17; 36]. For defense, adversarial training is meanwhile developed [35; 37; 38]. In particular, recent studies have shown that adversarial training can bring many benefits to GANs, not just improving robustness. RGAN [39] shows that adversarial training can enhance the generalization of the generator and discriminator. Based on theoretical analysis and empirical results, [40; 41; 42] show that updating the discriminator's parameters with the loss calculated by adversarial disturbed image samples can stabilize training and accelerate convergence.

In this paper, we use an off-the-shelf adversarial sample mining method *iterative fast gradient sign method (I-FGSM)* [17], to perform latent transform and mine effective $\mathbf{z}$ samples that benefit GAN training in its bi-level optimization process. More adversarial methods e.g. PGD [36] and MI-FGSM [43] are also tried and discussed in Appendix D. The pre-existing works mentioned above that interplay between GANs and adversarial learning make perturbations on raw pixel space (real samples or generated samples), while we instead work on mining the latent space and introducing perturbations on latent samples, which we show is a novel and effective way to improve the generation.

## 3 Proposed Method

This section presents our twofold effort to lift the final generative performance: AdvLatGAN-z to achieve the targeted transform $z^*(\cdot)$; training algorithms AdvLatGAN-qua and AdvLatGAN-div to train generation mapping $G$ with better generation quality (i.e., -qua) and diversity (i.e., -div). Both methods are derived by introducing the iterative updating in latent space.

### 3.1 Preliminaries

This section introduces adversarial attack/defense techniques and the regularization technique devised in MSGAN [32] for mode seeking. Adversarial techniques include: i) adversarial samples which we introduce in the latent space sampling to achieve $z^*(\cdot)$; ii) adversarial training which we introduce in the bi-level optimization of GAN to develop new training algorithms for a more powerful $G(\cdot)$.

**Adversarial samples and adversarial training.** Adversarial training [36; 44] aims to defend against adversarial attacks. It uses adversarial samples to calculate the loss. In classification, for each image-label pair $(\mathbf{x}_i, y_i)$ from the labeled training set, the adversarial samples $\mathbf{x}'$ [34] is defined as:

$$\mathbf{x}'_i = \underset{||\mathbf{x}-\mathbf{x_i}||_p \leq \epsilon}{\arg\max} \; l(f_\theta(\mathbf{x}), y_i) \tag{1}$$

Here $\epsilon$ is the radius of the closed ball to constrain perturbation. While $f_\theta$ and $l$ denote the attack target (a neural net) and the corresponding loss (e.g. cross-entropy loss). $p$ denotes the norm dimension. Given in total $n$ adversarial samples, adversarial training is fulfilled by the optimization:

$$\theta = \underset{\theta}{\arg\min} \; \frac{1}{n} \sum_{i=1}^{n} l(f_\theta(\mathbf{x}'_i), y_i) \tag{2}$$

**Mode coverage by regularizing distance of generated samples.** MSGAN [32] mitigates mode collapse for conditional generation by adding a regularizer over randomly sampled $\mathbf{z}_1$ and $\mathbf{z}_2$:

$$\mathcal{L}_{ms} = \frac{d_I\left(G(c, \mathbf{z}_1), G(c, \mathbf{z}_2)\right)}{d_z\left(\mathbf{z}_1, \mathbf{z}_2\right)} \tag{3}$$

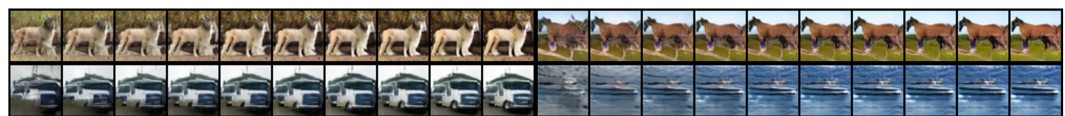

Figure 4: Visualization results of AdvLatGAN-z on STL-10. The left column shows the generations of raw Gaussian samples, while samples from left to right present the process of AdvLatGAN-z.

where $c$ is the condition vector for generation, and $d_I$ and $d_z$ are the distance metrics in the target (image) space and latent space, respectively. It encourages the distance to be maintained or even enlarged in the target space to avoid the generated samples being trapped into an aggregation.

By integrating the regularization term with the original objective of cGANs [45], the final minimization objective in MSGAN for training $G$ and $D$ is:

$$\mathcal{L} = \mathcal{L}_{ori} + \lambda_{ms}\mathcal{L}_{ms}, \quad \mathcal{L}_{ori} = \mathbb{E}_{c,\mathbf{x}}[\log D(c,\mathbf{x})] + \mathbb{E}_{c,\mathbf{z}}[\log(1 - D(c, G(c,\mathbf{z})))] \tag{4}$$

where $\mathcal{L}_{ori}$ denotes the original objective, $\mathbf{x}$ denotes real data, and $\lambda_{ms}$ is the weight hyper-parameter. Since Eq. 3 is to be maximized hence one can interpret $\lambda_{ms}$ as a negative coefficient.

## 3.2 Latent Adversarial Mining for Generation Quality

### 3.2.1 AdvLatGAN-z: Latent Space Transform

Note that as the real-data space is a union of disconnected manifolds [15], via a continuous generative mapping $G$, the optimal subset in the latent space w.r.t the fixed $G$ is in no way connected. We denote the real-data subset in the high-dimensional pixel space $X$ as $X_r \subset X$, and each disjoint submanifold as $M_i$. Then we obtain $X_r = \bigcup_{i=1}^{n_r} M_i$ where $M_1$ to $M_{n_r}$ are disconnected to each other. Here we require the splitting of $X_r$ to the extent that each submanifold is itself a connected set.

**Definition 3.1.** Suppose $G[\cdot] : \mathcal{P}(Z) \to \mathcal{P}(X)$ is the image-of-set function corresponding to the generative mapping function $G(\cdot) : Z \to X$, where $Z$ and $X$ denote subsets in latent space and pixel space, and $\mathcal{P}(\cdot)$ denotes the power set. Then $Z^{op}(G)$ is the optimal subset corresponding to $G$ in latent space if $G[Z^{op}(G)] = X_r$.

**Proposition 3.2.** *For any given continuous $G[\cdot]$ whose codomain includes multiple disconnected manifolds, $Z^{op}(G)$ is a union of disconnected subsets in the latent space.*

The proof is given in Appendix B. Proposition 3.2 implies that with the fixed continuous $G$, fitting the real-data manifolds requires a disconnected latent space support. Below we show how to implicitly transform the latent raw samples from Gaussian (or other forms e.g. uniform) to a new distribution closer to $Z^{op}(G)$ via the idea of mining adversarial samples as introduced above.

We utilize the discriminator's output to quantify the generation quality, which is trained to capture the difference between the generated and real distribution. Considering that small perturbations on $z$ can cause considerable quality variation (technically $D$'s output) as shown in Fig. 1 and Fig. 3, we naturally resort to adversarial perturbation-based methods to achieve the transform. Guided by Eq. 1, we first deliver the constrained optimization below, where both $G$ and $D$ are post-trained.

$$\mathbf{z}^*(\mathbf{z}_0) = \underset{\mathbf{z} \in \{\mathbf{z}|d(\mathbf{z}_0,\mathbf{z})\leq\epsilon\}}{\arg\min} \log(1 - D(G(\mathbf{z}))) \tag{5}$$

where $\mathbf{z}_0$ denotes the original latent vector, $\mathbf{z}^*(\mathbf{z}_0)$ denotes the newly computed $\mathbf{z}$, $d(\cdot,\cdot)$ denotes the distance which is $\ell_\infty$ in this paper, in line with the popular protocol in literature [35].

The above target can be readily solved by the well-developed tools in adversarial sample mining, e.g. the classic iterative fast gradient sign method (I-FGSM) [17] as formulated below in Eq. 6. In Appendix D, we also present comparison of more adversarial techniques e.g. PGD [36], MI-FGSM [43], that can be seamlessly incorporated by our framework, and we show the simple I-FGSM outperforms. We conjecture the reason may be due to the simplicity of the latent space in our case.

$$\mathbf{z}_{i+1} \leftarrow \mathbf{z}_i - \epsilon \cdot sgn\left(\nabla_{\mathbf{z}_i}\log(1 - D(G(\mathbf{z}_i)))\right) \tag{6}$$

From the distribution perspective, the transform is given by:

$$p_z^* = \underset{D(p_z,p_z^0)\leq\epsilon}{\arg\min} E_{\mathbf{z}\sim p_z}\left[\log(1 - D(G(\mathbf{z})))\right] \tag{7}$$

where $p_z^0$ denotes the original latent distribution, $p_z^*$ denotes the newly searched one. $D(p_z, p_z^0) \leq \epsilon$ means that there is a limit to the distribution variation. A newly mined $\mathbf{z}$ can be considered as a sample from $p_z^*$ if the sampling is within the perturbation circle. Fig. 4 shows the results by the pre-trained SNGAN [8] (i.e. the given $G$ and $D$) on STL-10 [46].

We then show that Eq. 7 guarantees the correct optimization direction that optimizes the source distribution $p_z$ towards the optimal probability distribution in latent space according to the current fixed $G$, i.e. $p_z^{op}(G)$ defined in Definition. 3.3. The proof for Proposition 3.4 is given in Appendix B.

**Definition 3.3.** Given $G$, $p_z^{op}(G)$ is the optimal probability distribution in latent space corresponding to $Z^{op}(G)$ when $p_z^{op}(G)$ satisfies that if $z \sim p_z^{op}(G)$ then $G(z) \sim p_r$. Here $p_r$ denotes the real-data probability distribution in pixel space.

**Proposition 3.4.** *Optimizing GAN's training criterion i.e.* $\min_G \max_D \{ E_{\mathbf{z} \sim p_z^0}[\log(1 - D(G(\mathbf{z})))] + E_{\mathbf{x} \sim p_r}[\log(D(\mathbf{x}))] \}$ *is to minimize* $JSD(p_z^{op}(G), p_z^0)$.

Prop. 3.4 implies that the GAN's optimization is to minimize the difference between $p_z^{op}(G)$ and $p_z^0$. Note that for optimizing $p_z$, the criterion merely involves $E_{\mathbf{z} \sim p_z}[\log(1 - D(G(\mathbf{z})))]$ i.e. the objective in Eq. 7. Given optimized fixed $G$ and $D$ (and $p_z^{op}$ is thereby fixed), leaving $p_z$ the only trainable parameter, optimizing the training criterion with $p_z$ in the neighborhood of $p_z^0$ is driving $p_z$ closer to $p_z^{op}$. Note that $p_z$ in the neighborhood of $p_z^0$ is necessary otherwise the optimized criterion will depart much from JSD, and we achieve this condition by enforcing $D(p_z, p_z^0) \leq \epsilon$ in Eq. 7.

### 3.2.2 AdvLatGAN-qua: Quality Targeted Training Pipeline

Considering the pairwise approach for adversarial sample mining, i.e. adversarial training fulfilled by Eq. 2, we hope to mine samples that benefit the optimization in GAN training, which is achieved by utilizing the aforementioned latent space transform. Conducting the transform in training to obtain the new (probably disconnected) latent distribution for each iteration can compensate for the difficulty for generators to align the pace of generators and discriminators, as it is widely recognized that it is much more challenging to train the generator than the discriminator, for its inherent defect that it cannot map a continuous distribution to a disconnected one with multiple modes.

Specifically, the transform is conducted to the raw Gaussian sampling in $D$ updating iterations while the rest of the training algorithm remains consistent with vanilla GAN [1] and other variants with the orthogonal efforts to latent space mining. We present the algorithm in Alg. 1 in the appendix which we call Adversarial Latent GAN with a post meaning quality (AdvLatGAN-qua). The latent transform is first computed by I-FGSM to mine the targeted samples in latent space, then $D$ and $G$ are updated. The iteration continues until enough steps or the convergence of $G$.

The above latent transform model does not consider how to avoid mode collapse, as each time only one latent sample is considered for optimization in isolation. A similar constrained optimization formulation can be devised to improve generation diversity, as shown in the following subsection.

### 3.3 AdvLatGAN-div: Latent Adversarial Mining for Diversity

In this section, we tend to modify the regularizer for mode seeking in MSGAN [32] by substituting the randomly selected regularized $\mathbf{z}$ pairs with hard sample pairs that are more inclined to collapse, lifting the efficiency for avoiding mode collapse. We first show how to search for such hard sample pairs. Given a sample $\mathbf{z}_0$ from the raw latent distribution, we aim to search its paired sample $\mathbf{z}^*$ to form a hard sample pair, in the sense that their generations are close in the target space given $G$. We consider two aspects. First, as hard samples, the paired samples shall still belong to the same category $\mathbf{c}$ which can be modeled with conditional GAN. Second, their distance in the latent space shall also be not too close, otherwise they may be aggregated in the target space due to continuous mapping. In other words, hard samples causing mode collapse shall be those close in target space while apart in latent space, when $G$ is given. Hence, guided by Eq. 1, we design our search procedure as the form:

$$\mathbf{z}^*(\mathbf{z}_0) = \underset{\mathbf{z} \in \{\mathbf{z} | d(\mathbf{z}_0, \mathbf{z}) \leq \epsilon\}}{\arg\min} \frac{d_I(G(\mathbf{c}, \mathbf{z}_0), G(\mathbf{c}, \mathbf{z}))}{d_z(\mathbf{z}_0, \mathbf{z})} \tag{8}$$

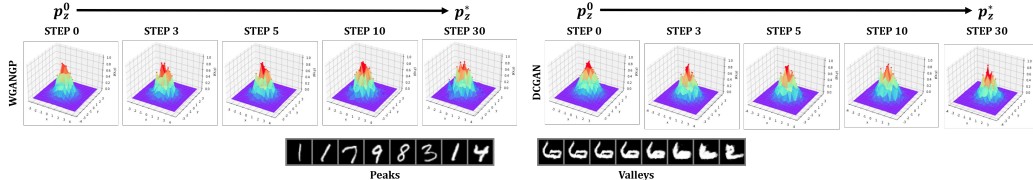

Figure 5: Latent distribution transform with WGAN-GP/DCGAN as backbones on MNIST. As the I-FGSM iteration guided by Eq. 5 continues, the latent distribution deviates from the standard Gaussian, and contains more valleys and peaks which can be mapped into generated images as shown at the bottom. Note that the generations by the valley points are of low quality, and our scheme can effectively avoid them as the sampling probability is low (valley) in our transformed distribution.

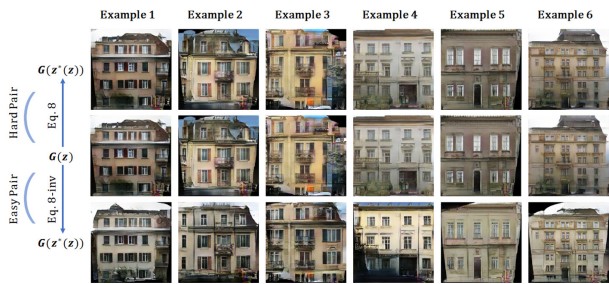

Figure 6: Each of the six columns shows two compared pairs of generated examples (first pair: 1st + 2nd row; second pair: 3rd row + 2nd row) by our diversity driven iterative transform scheme in latent space with the same number of iterations to obtain $z^*$ from initial $z$. **Middle:** generation by vanilla latent sampling $z$; **Top:** by latent samples mined by Eq. 8; **Bottom:** by latent samples mined by Eq. 8's inverse form. It shows that using Eq. 8 generates more similar image pairs serving as hard training samples.

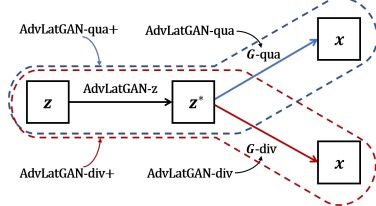

Figure 7: The logic of our 5 variant methods in Sec. 4.1. Blue and red is for quality and diversity.

Table 2: Five variants in Sec. 4.1.

| Variant | Latent Transform $z^*$ | Improved mapping $G$ |
|---|---|---|
| i) | ✓ | ✗ |
| ii) | ✗ | ✓ |
| iii) | ✗ | ✓ |
| iv) | ✓ | ✓ |
| v | ✓ | ✓ |

The symbols and terms are similar to Eq. 3. Akin to AdvLatGAN-qua, we use I-FGSM to obtain $\mathbf{z}^*$ and again use $\ell_\infty$ to control deviation magnitude. For each update step, we randomly select the first latent vector $\mathbf{z}_0$, and the second vector $\mathbf{z}^*$ is searched by $n$ iterations of the I-FGSM method under the $\ell_\infty$ restriction starting from the Gaussian neighborhood of $\mathbf{z}_0$ with a small standard deviation.

Table 1: Ratio between image and latent distance by Eq. 8 to derive new $\mathbf{z}^*$. *-inv*: using inverse form.

| Dataset | Eq. 8 | Eq. 8-inv |
|---|---|---|
| Cifar-10 | 0.241 | 4.023 |
| FACADES | 0.3785 | 1.3385 |

Table 1 shows the effectiveness of the above paired hard sample seeking method, where the results are calculated over 5k generated images with DCGAN [47] as backbone. The visual results on Facades as shown in Fig. 6 suggest that the pair obtained by solving Eq. 8 tend to collapse (as they are hard samples), while the opposite leads to more different generation by using the inverse form of Eq. 8.

The above experiments verify the efficacy of our mining technique. We further introduce it into the initial training process of MSGAN [32]. In MSGAN, the regularization term by Eq. 3 regularizes the optimization process, where the selection of the pair is entirely random, without using any heuristic information. We instead use the $\mathbf{z}$ pair that is more inclined to collapse by randomly taking one $\mathbf{z}$ and iterating by the aforementioned method to get another $\mathbf{z}$. The other settings remain unchanged.

By using sample pairs mined by Eq. 8 in the optimization as inspired by Eq. 2, we finally develop our new diversity enhancing GAN training approach AdvLatGAN-div as shown in Alg. 2 in the appendix, which can be further enhanced by post-training as discussed in experiments.

# 4 Experiments

Experiments on ImageNet and CelebA are performed on GPUs of Tesla V100. Other public benchmark results are performed on a single GPU of GeForce RTX 3090.

Table 3: Post-training latent sampling improvement with three architectures on STL-10 (best in **bold**).

| Method | DCGAN | | WGAN-GP | | SNGAN | |
|---|---|---|---|---|---|---|
| | IS($\uparrow$) | FID($\downarrow$) | IS($\uparrow$) | FID($\downarrow$) | IS($\uparrow$) | FID($\downarrow$) |
| bare w/o sampling transform | 7.199±0.027 | 58.675±0.023 | 8.749±0.093 | 35.706±0.395 | 8.482±0.138 | 37.059±0.768 |
| EvolGAN [25] | 7.174±0.086 | 58.761±0.030 | 8.773±0.068 | 35.851±0.003 | 8.517±0.069 | 36.343±0.512 |
| DOT [53] | 7.195±0.022 | 58.670±0.076 | 9.708±0.013 | 30.745±0.113 | 8.798±0.024 | 36.769±0.133 |
| DDLS [30] | 7.300±0.131 | 62.256±1.042 | 9.370±0.125 | 32.738±0.436 | 8.595±0.071 | 35.301±0.161 |
| AdvLatGAN-z | **7.933±0.041** | **54.913±0.451** | **10.549±0.048** | **27.053±0.245** | **9.426±0.058** | **31.891±0.027** |

## 4.1 Experimental Setup

We validate the effectiveness of the proposed methods for two parts, i.e., effective latent sampling improvement $z^*$ and improved generation mapping $G$. Five variant methods can be derived for different targets, among which AdvLatGAN-qua+ and AdvLatGAN-div+ are our final full version.

**i) AdvLatGAN-z:** post-training latent sampling improvement fighting against quality discontinuity; **ii) AdvLatGAN-qua:** GAN training algorithm for better quality using in-training latent sampling transform; **iii) AdvLatGAN-div:** GAN training algorithm for a more diverse generation by using in-training latent sampling transform; **iv) AdvLatGAN-qua+:** conducting -z over the trained networks of -qua; **v) AdvLatGAN-div+:** conducting -div over the trained networks of -div. We compare these methods in Fig. 7 and Table 2.

We adopt the Inception Score, IS [48], Fréchet Inception Distance, FID [49] and density/coverage [50] for evaluation. IS utilizes the classification results of the generated images by the Inception Network to evaluate the generation quality. FID additionally considers the target distribution and evaluates the quality by fitting the distribution distance. Density and coverage are more recently proposed evaluation metrics that analyze the target and generated manifolds and estimate the similarity. In the following experiments, IS and FID are calculated over 50,000 images, while density and coverage are calculated over 10,000 images. Overhead discussion is presented in Appendix C.

## 4.2 AdvLatGAN-z: Performance Boosting by Post-training Latent Sampling

We evaluate the boosting effect of post-training latent sampling improvement method AdvLatGAN-z (also previously denoted as $z^*$), which is guided by Eq. 5. It aims to mitigate quality discontinuity.

**Results on Synthetic Data.** We test AdvLatGAN-z on Grid and Ring datasets consisting of a mixture of 25 and 8 2-D Gaussians as done in [51]. The result is given in Fig. 9. By updating the latent variables, our method achieves high-quality generation (samples closer to target Gaussian centers).

**Results on MNIST.** Low complexity of MNIST [52] makes it possible to generate recognizable results by two-dimensional latent variables which can be better visualized. The iterative latent distribution change is illustrated in Fig. 5. We respectively conduct AdvLatGAN-z for the trained DCGAN and WGAN-GP models, with single-step iteration constraint using $\ell_\infty$ and iterated step size $\epsilon$ is set as 0.03. We present the first 30 iterations that can reflect the most significant changes.

**Results on STL-10.** STL-10 [46] maintains a higher resolution and richer image representation. We utilize AdvLatGAN-z for pre-trained DCGAN, WGAN-GP and SNGAN, with $\ell_\infty$ single-step constraint and the iteration step size $\epsilon$ is set as 0.05. We conduct 20 steps each time. IS and FID are adopted for evaluation, which are calculated over 10,000 generated images. We compare other efforts enhancing GAN performance by latent space mining including DOT, DDLS and EvolGAN. Table 3 present the results. AdvLatGAN-z outperforms the baselines and delivers at most 24.3% advantage for FID on vanilla in the WGAN-GP setting.

**Results on AFHQ and FFHQ.** To address the issues in Fig. 3, we evaluate on AFHQ [54] and FFHQ [55] based on StyleGAN2-ada [18]. In Fig. 3 setting, when the sampling encounters invalid examples, AdvLatGAN-z can transform the bad $z$ to generation-friendly $z^*(z)$ to avoid bad generation. Here we verify whether invalid samples can be improved with AdvLatGAN-z. We select six bad generations with the first three corresponding to those in Fig. 3 and conduct AdvLatGAN-z to their latent vectors. Fig. 10 shows that AdvLatGAN-z can effectively fix or avoid defects.

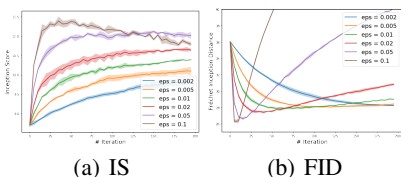

(a) IS (b) FID

Figure 8: Influence of step size $\epsilon$ and #$Iteration$: mean and std of different random seeds.

Table 4: Results of AdvLatGAN-qua and AdvLatGAN-qua+ for Cifar-10 and STL-10 (best in **bold**).

| Dataset | Method | bare | | AdvLatGAN-qua | | AdvLatGAN-qua+ | |
|---|---|---|---|---|---|---|---|
| | | IS($\uparrow$) | FID($\downarrow$) | IS($\uparrow$) | FID($\downarrow$) | IS($\uparrow$) | FID($\downarrow$) |
| Cifar-10 | DCGAN [47] | $5.92 \pm 0.05$ | $46.4 \pm 0.7$ | $6.21 \pm 0.07$ | $41.7 \pm 1.0$ | $\mathbf{6.62 \pm 0.26}$ | $\mathbf{40.2 \pm 1.9}$ |
| | WGAN [7] | $6.63 \pm 0.05$ | $32.8 \pm 0.5$ | $7.21 \pm 0.04$ | $27.3 \pm 0.7$ | $\mathbf{7.93 \pm 0.02}$ | $\mathbf{26.9 \pm 0.6}$ |
| | WGAN-GP [56] | $7.47 \pm 0.09$ | $24.7 \pm 0.2$ | $7.60 \pm 0.06$ | $22.6 \pm 0.4$ | $\mathbf{8.84 \pm 0.08}$ | $\mathbf{16.6 \pm 0.4}$ |
| | SNGAN [8] | $7.29 \pm 0.09$ | $25.5 \pm 0.3$ | $7.58 \pm 0.03$ | $22.3 \pm 0.5$ | $\mathbf{8.33 \pm 0.03}$ | $\mathbf{18.1 \pm 0.1}$ |
| | LSGAN [9] | $5.87 \pm 0.14$ | $49.3 \pm 2.2$ | $6.13 \pm 0.27$ | $42.8 \pm 1.3$ | $\mathbf{6.55 \pm 0.10}$ | $\mathbf{42.3 \pm 1.3}$ |
| | WGAN-div [57] | $7.43 \pm 0.02$ | $23.8 \pm 0.3$ | $7.81 \pm 0.03$ | $20.6 \pm 0.4$ | $\mathbf{8.87 \pm 0.11}$ | $\mathbf{15.2 \pm 0.6}$ |
| | ACGAN [58] | $6.02 \pm 0.28$ | $59.5 \pm 1.5$ | $6.06 \pm 0.21$ | $\mathbf{53.7 \pm 1.6}$ | $\mathbf{6.24 \pm 0.16}$ | $53.9 \pm 0.5$ |
| STL-10 | DCGAN [47] | $7.18 \pm 0.09$ | $61.2 \pm 1.2$ | $7.33 \pm 0.07$ | $56.3 \pm 1.0$ | $\mathbf{7.98 \pm 0.02}$ | $\mathbf{52.8 \pm 0.4}$ |
| | WGAN [7] | $6.51 \pm 0.07$ | $73.0 \pm 0.2$ | $7.62 \pm 0.04$ | $\mathbf{51.0 \pm 0.6}$ | $\mathbf{8.57 \pm 0.04}$ | $52.1 \pm 2.9$ |
| | WGAN-GP [56] | $8.86 \pm 0.05$ | $37.4 \pm 0.4$ | $8.90 \pm 0.05$ | $34.2 \pm 0.9$ | $\mathbf{10.78 \pm 0.02}$ | $\mathbf{25.0 \pm 0.0}$ |
| | SNGAN [8] | $8.49 \pm 0.09$ | $36.8 \pm 0.4$ | $8.63 \pm 0.08$ | $34.5 \pm 0.21$ | $\mathbf{9.64 \pm 0.05}$ | $\mathbf{29.9 \pm 0.1}$ |
| | LSGAN [9] | $7.08 \pm 0.12$ | $62.9 \pm 2.2$ | $7.16 \pm 0.15$ | $58.5 \pm 1.3$ | $\mathbf{7.85 \pm 0.08}$ | $\mathbf{55.3 \pm 1.4}$ |
| | WGAN-div [57] | $8.82 \pm 0.02$ | $37.7 \pm 0.2$ | $9.00 \pm 0.01$ | $32.0 \pm 0.6$ | $\mathbf{10.98 \pm 0.28}$ | $\mathbf{23.0 \pm 1.3}$ |

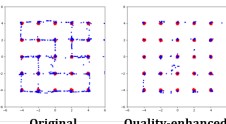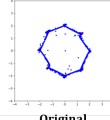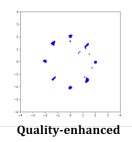

| Original | Quality-enhanced | Original | Quality-enhanced |
|---|---|---|---|

Figure 9: Post-training latent sampling improvement on Grid and Ring by AdvLatGAN-qua.

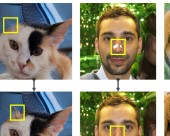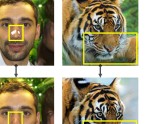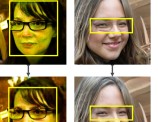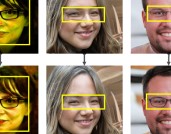

Figure 10: Results of AdvlatGAN-z on AFHQ and FFHQ. The top row are bad generations (the first three correspond to those in Fig. 3) with the defects as follows: **1st column**: the cat misses an ear; **2nd column**: white spots on the nose; **3rd column**: two bodies share one head; **4th column**: green face; **5th and 6th columns**: semi-existing glasses. The bottom row are results under AdvLatGAN-z. AdvLatGAN-z can effectively mitigate defects.

Table 5: Larger-scale evaluation (best in **bold**).

| Data | Methods | FID $\downarrow$ | Density $\uparrow$ | Coverage $\uparrow$ |
|---|---|---|---|---|
| LSUN-64 | WGANGP | $14.079 \pm 0.192$ | $0.747 \pm 0.013$ | $0.897 \pm 0.005$ |
| | WGANGP-qua | $12.415 \pm 0.028$ | $0.755 \pm 0.001$ | $0.909 \pm 0.003$ |
| | WGANGP-qua+ | $\mathbf{10.353 \pm 0.257}$ | $\mathbf{0.765 \pm 0.011}$ | $\mathbf{0.926 \pm 0.021}$ |
| | SNGAN | $11.961 \pm 0.331$ | $1.059 \pm 0.014$ | $0.925 \pm 0.003$ |
| | SNGAN-qua | $9.917 \pm 0.104$ | $\mathbf{1.078 \pm 0.033}$ | $0.939 \pm 0.002$ |
| | SNGAN-qua+ | $\mathbf{7.285 \pm 0.029}$ | $1.063 \pm 0.003$ | $\mathbf{0.963 \pm 0.002}$ |
| LSUN-128 | WGANGP | $14.180 \pm 0.583$ | $0.754 \pm 0.047$ | $0.835 \pm 0.007$ |
| | WGANGP-qua | $12.581 \pm 0.145$ | $0.851 \pm 0.014$ | $0.878 \pm 0.013$ |
| | WGANGP-qua+ | $\mathbf{11.093 \pm 0.210}$ | $\mathbf{0.869 \pm 0.004}$ | $\mathbf{0.914 \pm 0.004}$ |
| | SNGAN | $16.078 \pm 0.193$ | $0.946 \pm 0.035$ | $0.863 \pm 0.003$ |
| | SNGAN-qua | $14.277 \pm 0.035$ | $1.097 \pm 0.052$ | $0.869 \pm 0.004$ |
| | SNGAN-qua+ | $\mathbf{11.244 \pm 0.396}$ | $\mathbf{1.133 \pm 0.068}$ | $\mathbf{0.908 \pm 0.004}$ |
| CelebA-64 | WGANGP | $19.834 \pm 0.398$ | $0.247 \pm 0.017$ | $0.596 \pm 0.017$ |
| | WGANGP-qua | $18.972 \pm 0.174$ | $0.246 \pm 0.005$ | $0.602 \pm 0.005$ |
| | WGANGP-qua+ | $\mathbf{17.748 \pm 0.285}$ | $\mathbf{0.262 \pm 0.012}$ | $\mathbf{0.631 \pm 0.003}$ |
| | SNGAN | $19.490 \pm 0.473$ | $0.295 \pm 0.002$ | $0.612 \pm 0.008$ |
| | SNGAN-qua | $18.255 \pm 0.209$ | $\mathbf{0.301 \pm 0.006}$ | $0.632 \pm 0.002$ |
| | SNGAN-qua+ | $\mathbf{17.213 \pm 0.104}$ | $0.279 \pm 0.007$ | $\mathbf{0.639 \pm 0.008}$ |
| CelebA-128 | WGANGP | $25.208 \pm 0.409$ | $0.204 \pm 0.005$ | $0.505 \pm 0.015$ |
| | WGANGP-qua | $22.376 \pm 0.181$ | $0.247 \pm 0.007$ | $0.579 \pm 0.012$ |
| | WGANGP-qua+ | $\mathbf{21.374 \pm 0.361}$ | $\mathbf{0.256 \pm 0.012}$ | $\mathbf{0.597 \pm 0.008}$ |
| | SNGAN | $22.940 \pm 0.092$ | $\mathbf{0.467 \pm 0.004}$ | $0.649 \pm 0.004$ |
| | SNGAN-qua | $22.476 \pm 0.023$ | $0.462 \pm 0.006$ | $0.651 \pm 0.005$ |
| | SNGAN-qua+ | $\mathbf{20.235 \pm 0.402}$ | $0.414 \pm 0.008$ | $\mathbf{0.668 \pm 0.005}$ |
| ImageNet | WGANGP | $78.150 \pm 0.298$ | $0.200 \pm 0.002$ | $0.289 \pm 0.005$ |
| | WGANGP-qua | $73.677 \pm 0.576$ | $0.242 \pm 0.005$ | $0.322 \pm 0.010$ |
| | WGANGP-qua+ | $\mathbf{71.227 \pm 0.444}$ | $\mathbf{0.251 \pm 0.002}$ | $\mathbf{0.333 \pm 0.004}$ |
| | SNGAN | $98.344 \pm 0.758$ | $0.188 \pm 0.003$ | $0.213 \pm 0.007$ |
| | SNGAN-qua | $79.087 \pm 0.546$ | $0.207 \pm 0.003$ | $0.266 \pm 0.007$ |
| | SNGAN-qua+ | $\mathbf{74.643 \pm 0.844}$ | $\mathbf{0.209 \pm 0.009}$ | $\mathbf{0.288 \pm 0.009}$ |

**Hyper-parameter Study.** Considering the influence of the step size $\epsilon$ and the number of iterations, we evaluate WGAN-GP's performance on STL-10 adopting IS and FID metrics. Fig. 8 shows the generative performance after the iteration number. We observe that a larger $\epsilon$ (e.g. 0.05) can deliver fast improvement but can be unstable for more iterations, while a smaller $\epsilon$ can achieve stable improvement but is slightly less significant. If time permits for sufficient parameter tuning, we recommend adopting a larger $\epsilon$ like 0.05 along with a particularly selected iteration number, otherwise a smaller $\epsilon$ like 0.01 with sufficient iterations will be enough for worthy performance gain.

## 4.3 AdvLatGAN-qua/div: Improving Generation Map

This section shows the effectiveness of our training pipelines, which aim to improve the generation mapping $G$ respectively for quality and diversity. The details are presented in Alg.1 and Alg.2.

**AdvLatGAN-qua for Quality Improvement.** Table 4 shows the results on Cifar-10 and STL-10, using the mainstream architectures: DCGAN, WGAN, WGAN-GP, SNGAN, LSGAN, WGAN-div and ACGAN. We do not include ACGAN in the unlabeled STL-10 setting because it requires labels [58]. IS and FID are adopted. We adversarially train GAN using AdvLatGAN-qua. The

Table 6: Results of AdvLatGAN-div and AdvLatGAN-div+ for Cifar-10 and STL-10 (best in **bold**).

| Dataset | Metrics | Models | overall | airplane | automobile | bird | cat | deer | dog | frog | horse | ship | truck |
|---|---|---|---|---|---|---|---|---|---|---|---|---|---|
| Cifar-10 | FID(↓) | MSGAN | 30.225 | 73.083 | 69.518 | 78.258 | 74.525 | 57.778 | 86.831 | 63.287 | 69.705 | 69.994 | 66.434 |
| | | AdvLatGAN-div | 27.054 | 70.933 | 71.627 | 75.650 | 67.670 | **54.457** | **84.274** | 55.836 | 65.083 | 69.662 | **63.127** |
| | | AdvLatGAN-div+ | **26.169** | **67.100** | **67.499** | **73.449** | **67.070** | 55.817 | 85.815 | **55.227** | **64.159** | **67.265** | 63.223 |
| | density(↑) | MSGAN | 0.517 | **0.351** | 0.226 | **0.289** | 0.605 | **0.536** | 0.264 | **0.685** | 0.307 | 0.353 | 0.221 |
| | | AdvLatGAN-div | **0.524** | 0.345 | 0.260 | 0.254 | **0.657** | 0.524 | **0.306** | 0.666 | **0.351** | 0.365 | 0.225 |
| | | AdvLatGAN-div+ | 0.492 | 0.348 | **0.311** | 0.227 | 0.595 | 0.430 | 0.296 | 0.590 | 0.332 | **0.366** | 0.247 |
| | coverage(↑) | MSGAN | 0.828 | 0.789 | 0.848 | 0.596 | 0.765 | 0.833 | **0.610** | 0.880 | **0.916** | 0.957 | 0.775 |
| | | AdvLatGAN-div | 0.844 | 0.850 | 0.908 | 0.600 | 0.793 | 0.864 | 0.607 | 0.919 | 0.915 | **0.962** | **0.788** |
| | | AdvLatGAN-div+ | **0.849** | **0.874** | **0.922** | **0.604** | **0.800** | **0.891** | 0.599 | **0.934** | 0.877 | 0.959 | 0.780 |
| STL-10 | FID(↓) | MSGAN | 67.849 | 92.021 | 125.723 | 108.434 | 118.938 | 111.784 | 133.680 | 140.486 | 121.907 | 101.232 | 101.059 |
| | | AdvLatGAN-div | 65.088 | 92.007 | 125.237 | 109.672 | 118.645 | 104.859 | 132.424 | 140.266 | 113.745 | 93.833 | 99.739 |
| | | AdvLatGAN-div+ | **60.205** | **88.871** | **118.886** | **104.206** | **111.238** | **99.255** | **125.287** | **131.170** | **106.568** | **91.895** | **95.715** |
| | density(↑) | MSGAN | 0.353 | 0.198 | 0.241 | 0.083 | 0.558 | 0.378 | 0.364 | 0.146 | 0.232 | 0.120 | 0.118 |
| | | AdvLatGAN-div | 0.429 | 0.220 | 0.292 | 0.114 | 0.686 | **0.404** | **0.384** | 0.193 | **0.336** | 0.164 | 0.123 |
| | | AdvLatGAN-div+ | **0.431** | **0.248** | **0.295** | **0.125** | **0.733** | 0.401 | 0.381 | **0.209** | 0.330 | **0.190** | **0.149** |
| | coverage(↑) | MSGAN | 0.455 | 0.538 | **0.535** | 0.261 | 0.539 | 0.380 | 0.391 | 0.370 | 0.410 | 0.456 | 0.324 |
| | | AdvLatGAN-div | 0.514 | 0.526 | 0.455 | 0.350 | 0.624 | 0.483 | 0.409 | 0.513 | 0.439 | 0.513 | 0.363 |
| | | AdvLatGAN-div+ | **0.540** | **0.554** | 0.519 | **0.364** | **0.699** | **0.514** | **0.433** | **0.550** | **0.484** | **0.564** | **0.383** |

number of latent iterations in training is set as one per discriminator step, and other details is given in Appendix J.1. As they are orthogonal to each other, we integrate AdvLatGAN-z with AdvLatGAN-qua (i.e. AdvLatGAN-qua+) and include it in comparison. The iteration step size $\epsilon$ of AdvLatGAN-z is set to 0.05 and we conduct 20 steps each time. Our method achieves a notable improvement compared to that without the latent space transform strategy. AdvLatGAN-qua+ achieves the best gain on IS in WGAN STL-10 setting from 6.51 to 8.57 and FID in WGAN-div from 37.7 to 23.0.

Table. 5 presents results on larger datasets: LSUN Church [59], CelebA [60] and ImageNet [61], using WGAN-GP and SNGAN as backbones, evaluated by FID, density and coverage. We include AdvLatGAN-qua and -qua+ in the comparison. The number of latent iterations during training is set as 1 per discriminator step, and other details are given in Appendix J.1. The iteration step size $\epsilon$ of AdvLatGAN-z is set to 0.01 for SNGAN and 0.002 for WGANGP and we conduct 100 steps each time. AdvLatGAN-qua+ has achieved the best performance gain on FID in SNGAN LSUN-64 setting from 11.961 to 7.285.

**AdvLatGAN-div for Diversity Improvement.** We test on Cifar-10 and STL-10 for conditional generation and we adopt FID, density and coverage as the metrics, respectively for the entire set of generated images and for those in each label. We test AdvLatGAN-div and AdvLatGAN-div+ compared to the baseline MSGAN. The results are shown in Table 6. Our method outperforms MSGAN by all the three metrics. Note that the models are trained on the overall set and for each label and the optimization may not guarantee targeted enough guidance. AdvLatGAN-div+ achieves the best overall performance gain on FID from 30.225 to 26.169, density from 0.353 to 0.431 and coverage from 0.455 to 0.54.

# 5 Conclusion and Broader Impact

This work analyzes GANs from the continuous mapping perspective and notes that lifting the overall generative performance requires a twofold effort including latent distribution transform and mapping improvement. Adversarial sample mining techniques are introduced to explore latent space and novel training pipelines are derived to improve the generative mapping. The proposed AdvLatGAN has shown promising power in both quality and diversity. Our technique is basically orthogonal to existing mainstream methods. For its limitation, it remains open for how to effectively combine the proposed two techniques for diversity and quality. For potential negative social impact, fake content by the generation models including ours shall always be carefully treated to avoid abuse.

# Acknowledgements

This work was partly supported by National Key Research and Development Program of China (2020AAA0107600), National Natural Science Foundation of China (61972250, 72061127003), and Shanghai Municipal Science and Technology (Major) Project (2021SHZDZX0102,22511105100).

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
