# Appendix

## A  Detailed Algorithm for AdvLatGAN-qua and AdvLatGAN-div

The algorithms for AdvLatGAN-qua in Section. 3.2 and AdvLatGAN-div in Section. 3.3 are presented in Alg. 1 and Alg. 2 .

---

**Algorithm 1 AdvLatGAN-qua** $t_z, t_D$: the number of steps, other symbols are consistent with vanilla GAN.

---

  **Input:** $p_z$ e.g. Gaussian, $p_r$ e.g. real images distribution, randomly initialized $G$ and $D$
  **Output:** trained generator $G$ and discriminator $D$
  **while** $G$ has not converged **do**
     *// update D:*
     **for** $i = 1$ to $t_D$ **do**
        Sample $\{\mathbf{z}_0^{(k)}\}_{k=1}^m \sim p_z$;
        *// the proposed sampling shift (t-step I-FGSM):*
        **for** $i = 1$ to $t_z$ **do**
           Obtain $\{\mathbf{z}_i^{(k)}\}_{k=1}^m$ by Eq. 5;
        **end for**
        Sample $\{\mathbf{x}_i^{(k)}\}_{k=1}^m \sim p_r$;
        Calculate $loss_D$ with $\{\mathbf{x}_i^{(k)}\}_{k=1}^m$ and $\{\mathbf{z}_{t_z}^{(k)}\}_{k=1}^m$;
        Update $D$ with $loss_D$;
     **end for**
     *// update G:*
     Sample $\{\mathbf{z}^{(k)}\}_{k=1}^m \sim p_z$;
     Calculate $loss_G$ with $\{\mathbf{z}^{(k)}\}_{k=1}^m$;
     Update $G$ with $loss_G$;
  **end while**

---

**Algorithm 2 AdvLatGAN-div** $t_z, t_D$: the number of steps, other symbols are consistent with vanilla GAN.

---

  **Input:** $p_z$ e.g. Gaussian, $p_r$ e.g. real images distribution; $\lambda_{ms}$ is a hyper-parameter; randomly initialized $G$ and $D$;
  **Output:** trained generator $G$ and discriminator $D$;
 1: **while** $G$ has not converged **do**
 2:     *// update D:*
 3:     **for** $i = 1$ to $t_D$ **do**
 4:         Sample $\{\mathbf{z}_i^{(k)}\}_{k=1}^m \sim p_z$;
 5:         Sample $\{(\mathbf{x}_i^{(k)}, c_i^{(k)})\}_{k=1}^m \sim p_r$;
 6:         Compute $loss_D$ with $\{(\mathbf{x}_i^{(k)}, c_i^{(k)}), \mathbf{z}_i^{(k)}\}_{k=1}^m$;
 7:         Update $D$ with $loss_D$;
 8:     **end for**
 9:     *// update G:*
10:     Randomly generate labels $\{c^{(k)}\}_{k=1}^m$;
11:     Sample $\{\mathbf{z}_0^{(k)}\}_{k=1}^m \sim p_z$;
12:     *// the proposed sampling shift (t-step I-FGSM):*
13:     **for** $i = 1$ to $t_z$ **do**
14:         Obtain $\{\mathbf{z}_i^{(k)}\}_{k=1}^m$ by Eq. 8;
15:     **end for**
16:     Calculate $loss_G$ and $loss_{ms}$ (Eq. 3) with $\{c^{(k)}\}_{k=1}^m$, $\{\mathbf{z}_0^{(k)}\}_{k=1}^m$ and $\{\mathbf{z}_{t_z}^{(k)}\}_{k=1}^m$;
17:     $loss \leftarrow loss_G + \lambda_{ms} loss_{ms}$;
18:     Update $G$ with $loss$;
19: **end while**

---

# B Proofs

We first recall the notations: $G(\cdot) : Z \to X$ denotes the generative mapping where $Z$ and $X$ denote subsets in latent space and pixel space; $G[\cdot] : \mathcal{P}(Z) \to \mathcal{P}(X)$ denotes the image-of-set function corresponding to $G(\cdot)$; $G^{-1}[\cdot] : \mathcal{P}(X) \to \mathcal{P}(Z)$ denotes the preimage function such that $G^{-1}[X] = \{z | G(z) \in X\}$, distinguished from the inverse function $G^{-1}(\cdot) : X \to Z$.

To prove Proposition. 3.2, we first show Lemma. B.1 for latter use.

**Lemma B.1.** *If $G : Z \to X$ is a continuous function, then for any open set $U$ in $X$, $G^{-1}[U]$ is an open set in $Z$.*

*Proof.* Let $\mathbf{z}_0 \in G^{-1}[U]$, then $G(\mathbf{z}_0) \in U$. Since $U$ is open, there exists a $\epsilon > 0$ such that $\{\mathbf{x} | d(\mathbf{x}, G(\mathbf{z}_0)) < \epsilon\} \subseteq U$. Here $d(\cdot, \cdot)$ denotes the Euclidean distance metric function. On the other hand, since $G$ is a continuous function, there exists a $\delta > 0$ such that for any $\mathbf{z} \in \{\mathbf{z} | d(\mathbf{z}, \mathbf{z}_0) < \delta\}$, $G(\mathbf{z}) \in \{\mathbf{x} | d(\mathbf{x}, G(\mathbf{z}_0)) < \epsilon\} \subseteq U$. So that for any $\mathbf{z} \in \{\mathbf{z} | d(\mathbf{z}, \mathbf{z}_0) < \delta\}$, $G(\mathbf{z}) \in U$ and $\mathbf{z} \in G^{-1}[U]$, i.e. $\{\mathbf{z} | d(\mathbf{z}, \mathbf{z}_0) < \delta\} \subseteq G^{-1}[U]$.

Concludingly put, for any $\mathbf{z}_0 \in G^{-1}[U]$, there exists a $\delta > 0$ such that $\{\mathbf{z} | d(\mathbf{z}, \mathbf{z}_0) < \delta\} \subseteq G^{-1}[U]$. According to the definition of open set, $G^{-1}[U]$ is open, completing the proof. $\square$

**Proposition B.2.** *For any given continuous $G$ whose codomain includes multiple manifolds, $Z^{op}(G)$ is a union of disconnected subsets in the latent space.*

*Proof.* Recall that the targeted real-data subset in the high-dimensional pixel space $X_r = \bigcup_{i=1}^{n_r} M_i$ where $M_1$ to $M_{n_r}$ are disconnected to each other and here we require the splitting of $X_r$ to such an extent that each submanifold $M_i$ keeps connected.

Since $M_1$ to $M_{n_r}$ are disconnected to each other and maintain connected themselves, there exist open sets $U_1$ to $U_{n_r}$ such that $M_i \subseteq U_i$ and $\forall i \neq j : U_i \bigcap U_j = \emptyset$.

Since $G[G^{-1}[M_i]] = M_i \subseteq U_i$, we obtain $G^{-1}[M_i] \subseteq G^{-1}[U_i]$. And according to Lemma. B.1, $G^{-1}[U_i]$ is open in $Z$. Then to show that $G^{-1}[U_i]$ to $G^{-1}[U_{n_r}]$ are disconnected to each other, we need to prove that $\forall i \neq j : G^{-1}[U_i] \bigcap G^{-1}[U_j] = \emptyset$.

We will aim for a contradiction. If there exists an element $\mathbf{a}$ such that $\mathbf{a} \in G^{-1}[U_i]$ and $\mathbf{a} \in G^{-1}[U_j]$, then $G(\mathbf{a}) \in U_i$ and $G(\mathbf{a}) \in U_j$. This implies that $G(\mathbf{a}) \in U_i \bigcap U_j$, which contradicts the condition that $U_i \bigcap U_j = \emptyset$.

Till now, we have shown that $G^{-1}[M_i] \subseteq G^{-1}[U_i]$ where $G^{-1}[U_i]$ is open in $Z$ and $\forall i \neq j : G^{-1}[U_i] \bigcap G^{-1}[U_j] = \emptyset$. This implies that $G^{-1}[M_i]$ to $G^{-1}[M_{n_r}]$ are disconnected to each other.

On the other hand, we have

$$Z^{op}(G) = G^{-1}[X_r] = G^{-1}[\bigcup_{i=1}^{n_r} M_i] = \bigcup_{i=1}^{n_r} G^{-1}[M_i]$$

Thus we obtain $Z^{op}(G)$ is a union of disconnected subsets in the latent space, completing the proof. $\square$

**Proposition B.3.** *Optimizing GAN's training criterion i.e. $\min_G \max_D \{E_{\mathbf{z} \sim p_z^0}[\log(1 - D(G(\mathbf{z})))] + E_{\mathbf{x} \sim p_r}[\log(D(\mathbf{x}))]\}$ is to minimize $JSD(p_z^{op}(G), p_z^0)$.*

*Proof.* We introduce $p_z^{op}(G)$ to vanilla GAN's training criterion and denote the target function as $V(G, D)$, then

$$
\begin{aligned}
&V(G, D) \\
=&E_{\mathbf{z} \sim p_z^0}[\log(1 - D(G(\mathbf{z})))] + E_{\mathbf{x} \sim p_r}[\log(D(\mathbf{x}))] \\
=&E_{\mathbf{z} \sim p_z^0}[\log(1 - D(G(\mathbf{z})))] + E_{\mathbf{z} \sim p_z^{op}}[\log(D(G(\mathbf{z})))] \\
=&\int_{\mathbf{z}} p_z^0(\mathbf{z}) \log(1 - D \circ G(\mathbf{z})) + p_z^{op}(\mathbf{z}) \log(D \circ G(\mathbf{z})) d\mathbf{z}
\end{aligned}
$$

Similar to the derivation of Thm. 1 in [1], we assume that with the fixed $G$, $D$ have reached optimal with a derivative of zero. Let $\frac{\partial V}{\partial D} = 0$, we obtain

$$D^* \circ G(\mathbf{z}) = \frac{p_z^{op}}{p_z^{op} + p_z^0} \tag{9}$$

Then the criterion turns to be $C(G)$:

$$
\begin{aligned}
&C(G)\\
=&KLD\left(p_z^{op}\|\frac{p_z^{op} + p_z^0}{2}\right) + KLD\left(p_z^0\|\frac{p_z^{op} + p_z^0}{2}\right) - 2\log 2\\
=&2JSD(p_z^{op}\|p_z^0) - 2\log 2
\end{aligned}
$$

where $KLD(\cdot)$ denotes the Kullback-Leibler divergence and $JSD(\cdot)$ denotes the Jensen-Shannon divergence. Since the JSD of two distributions is always non-negative and reaches zero if and only if the two distributions are the same, completing the proof.

$\square$

## C  Overhead Discussion

We conduct experiments to show the overhead of our proposed techniques on both the in-training (*-qua* as an example) and post-training sampling shift method (*-z*) on STL10. We evaluate training time for 1000 generator steps for *-qua* and time for shifting 1000 samples for *-z*, in seconds. The comparison involves baselines in Table.3, with the same parameters. We compare two version of *-z* w.r.t. #iteration (correspond to Fig. 8). The results show *-z*'s high computational efficiency.

Table 7: Results for *-qua*.

| Method | STL-32 | STL-64 |
|---|---|---|
| SNGAN | $71.01 \pm 0.32$ | $526.52 \pm 0.51$ |
| SNGAN-qua | $95.96 \pm 0.26$ | $685.83 \pm 1.06$ |
| WGANGP | $103.20 \pm 0.25$ | $794.48 \pm 0.24$ |
| WGANGP-qua | $127.39 \pm 0.20$ | $955.02 \pm 0.37$ |

Table 8: Results for *-z*.

| Method | SNGAN | WGANGP |
|---|---|---|
| EvolGAN | $628.70 \pm 1.76$ | $630.55 \pm 1.61$ |
| DOT | $68.20 \pm 0.10$ | $69.78 \pm 0.08$ |
| DDLS | $267.41 \pm 34.99$ | $749.91 \pm 185.91$ |
| -z (it=100) | $34.12 \pm 0.11$ | $42.03 \pm 0.42$ |
| -z (it=20) | $6.96 \pm 0.26$ | $8.40 \pm 0.27$ |

## D  Relation to Adversarial Attacks/Defenses

Our approach is closely related to adversarial attacks and defenses. As for works in adversarial attacks, the fundamental standpoint is that deep neural networks' performance can vary significantly in face of small perturbations. This scenario can be very similar to our starting point as described in Sec 1 that a small perturbation of the latent vectors can lead to massive quality variation. Thus it is quite natural to combine the adversarial methods with generative adversarial networks, and specifically, AdvLatGAN-z achieves the implicit latent transform using I-FGSM to overcome the quality discontinuity.

The proposed GAN training pipelines AdvLatGAN-qua and AdvLatGAN-div can be analogous to adversarial training techniques in the adversarial defense field. Adversarial training is a proactive defense approach that strengthens the model against attacks or enhances its performance by modifying the inputs to adversarial samples to train the model, making it naturally robust and defensive against attacks. This process involves conducting adversarial sample mining during training which can be optimization-friendly for the robustness purpose. While in AdvLatGAN-qua and AdvLatGAN-div, we conduct $\mathbf{z}$ transform in GAN training referencing adversarial sample mining, which can be viewed as the same process as adversarial training. The training involves optimization of three components $\mathbf{z}$, $G$ and $D$, among which the introduced $\mathbf{z}$ transform can mine the latent space during training, to some extent leading to the superiority of the method.

**Adopting different adversarial perturbation methods.** Though we adopt I-FGSM for its simpleness, our framework is agnostic to the choice of adversarial techniques. We try and compare other means for adversarial mining, and below we give experiments using other adversarial methods including PGD [62], MI-FGSM [43]. We involve backbones as WGAN-GP and SNGAN, adopting IS and FID as the evaluation metrics. The results are presented in Table. 9 and Table. 10. I-FGSM

marginally outperforms other methods. This may stem from that the low dimensional latent space has a simpler structure than image space.

<table>
<tr><td colspan="4" align="center">Table 9: Results for -qua.</td></tr>
<tr><td>Method</td><td>Metric</td><td>WGAN-GP</td><td>SNGAN</td></tr>
<tr><td rowspan="2">I-FGSM</td><td>IS↑</td><td>7.63 ± 0.09</td><td>7.56 ± 0.07</td></tr>
<tr><td>FID↓</td><td>22.2 ± 0.3</td><td>21.9 ± 0.3</td></tr>
<tr><td rowspan="2">MI-FGSM</td><td>IS↑</td><td>7.63 ± 0.03</td><td>7.47 ± 0.05</td></tr>
<tr><td>FID↓</td><td>22.9 ± 0.2</td><td>22.4 ± 0.6</td></tr>
<tr><td rowspan="2">PGD</td><td>IS↑</td><td>7.63 ± 0.06</td><td>7.50 ± 0.07</td></tr>
<tr><td>FID↓</td><td>22.5 ± 0.2</td><td>22.2 ± 0.4</td></tr>
</table>

<table>
<tr><td colspan="4" align="center">Table 10: Results for -z.</td></tr>
<tr><td></td><td></td><td>WGAN-GP</td><td>SNGAN</td></tr>
<tr><td rowspan="2">I-FGSM</td><td>IS↑</td><td>8.84 ± 0.08</td><td>8.33 ± 0.03</td></tr>
<tr><td>FID↓</td><td>16.6 ± 0.4</td><td>18.1 ± 0.1</td></tr>
<tr><td rowspan="2">MI-FGSM</td><td>IS↑</td><td>8.71 ± 0.07</td><td>8.26 ± 0.01</td></tr>
<tr><td>FID↓</td><td>16.6 ± 0.3</td><td>18.2 ± 0.1</td></tr>
<tr><td rowspan="2">PGD</td><td>IS↑</td><td>8.56 ± 0.07</td><td>8.11 ± 0.01</td></tr>
<tr><td>FID↓</td><td>16.7 ± 0.2</td><td>18.8 ± 0.1</td></tr>
</table>

# E    Description of Constraints for Iteration

There are two basic types of constraint used in adversarial attacks: 1) $\ell_2$ norm constraint; 2) $\ell_\infty$ norm constraint. In this paper, we follow the well-known attack method, iterative fast gradient sign method (I-FGSM) [17] to conduct the iterations of latent variables, adopting the $\ell_\infty$ norm as our constraint.

# F    Universality of Fig. 3's phenomenon.

StyleGAN2-ada is just an example, and the issue exists for any network-based generator due to the continuous nature of net-based mappings as claimed in the main paper. Yellow box below shows similar artifacts phenomenon in BigGAN [12].

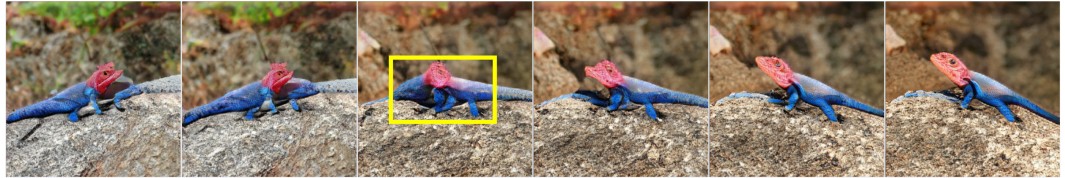

Figure 11: BigGAN results corresponding to Fig. 3.

# G    Comparisons to Other Algorithms

We present the differences between our approaches and other previous works, involving two topics i.e. latent exploration and GAN model with adversarial training. An overview is presented in Table 11. Meanwhile, we conduct experiments to show our superiority over other representative works.

## G.1    Latent Exploration

**EvolGAN [25], Tarsier [26].** These two models both use the well-known quality evaluator Koncept512 (which is actually a classifier) to guide the latent vector optimization. Their drawback is that they do not consider or exploit the real distribution but turn to a general quality estimator for optimization guidance, which leads to the criterion of sampling quality not for the proximity to the real distribution, but to match the data trained for the Koncept512 classifier. Our approach uses a discriminator to guide the latent variable updating, which makes full use of the information from the real distribution, while further achieving novel training algorithm to improve the generative mapping.

**DDLS [30].** DDLS analyzes GAN and develops the sample-improving approach from the perspective of the energy-based model, achieved by Markov Chain Monte Carlo, which differs from ours in terms of methodology. Besides, it does not particularly notice that when we enhance the quality, we hope that the image is changed as little as possible. Combing the above two points, we explain the problem from the perspective of continuous mapping and view the latent vector transform as adversarial sample mining, which is quite different from DDLS.

**DOT [53].** Mining the residual value of the discriminator after training, DOT proposes optimal transport in the pixel space and latent space to improve the generated images. The motivation of

Table 11: Differences from Other Algorithms

| Proposed Method | Compared Existing Method | Different Point | Detailed Description | |
|---|---|---|---|---|
| | | | Theirs | Ours |
| -z | EvolGAN | Guidance for the transform | Quality estimator Koncept512 | $D \cdot G$ |
| | | Technical implementation | Evolutionary algorithm | I-FGSM iterations |
| | Tarsier | Guidance of the transform | Quality estimator Koncept512 | $D \cdot G$ |
| | | Technical implementation | Diagonal Covariance Matrix Adaptation | I-FGSM iterations |
| | | Targeted Task | Super-resolution generation tasks | General generation tasks |
| | AE-OT-GAN | Timing for the transform | Before training G and D | After training G and D |
| | | Technical implementation | First train an Auto-Encoder to learn a latent distribution then use optimal trasport to achieve the transform | I-FGSM iterations |
| | | Calculating cost | Need additionally train an Auto-Encoder for fitting the targeted latent distribution | No extra networks are trained (guided by G and D) |
| | DDLS | Basis of theoretical analysis | Energy-based theory | General GAN theory |
| | | Technical implementation | Markov Chain Monte Carlo | I-FGSM iterations |
| | DOT | Motivation | Mining D's residual value after training | Quality discontinuity |
| | | Technical implementation | Optimal transport | I-FGSM iterations |
| -qua | vanilla GAN | $z$ in the training of D | Samples from Gaussian | Transform of the original sampling |
| -div | MSGAN | $z$ pair used in ms-regularizer | Individually sampled from Gaussian | Randomly choose $z_1$, then transform $z_1$ to get $z_2$, forming a hard sample pair |
| -qua and -div | Existing methods introducing AT or robustness learning in GAN training | Objects to which perturbations are added | $x$ | $z$ |

further mining the discriminator after training differs from ours to overcome the quality discontinuity, while again the technical implementation being optimal transport differs from ours.

## G.2    GAN Model with Adversarial Training

Our GAN training algorithm can be considered as a kind of adversarial training, and we differ from several previous works that combine adversarial training and GAN.

**ASGAN [42], FastGAN [41], Rob-GAN [40].** These methods add perturbations to the real images (the latter is an improvement in the loss function compared to the former), while we focus on the latent space, perturbing latent space vectors.

**Robust GAN training [63].** This algorithm shows that a robust discriminator can benefit training and this robustness only need to be enforced in expectation over the generated samples, again without focusing on the latent space compared to us.

Table 12: Experimental comparisons over GAN with adversarial training evaluated on CIFAR-10.

| Metrics | Framework | ASGAN | RobDis | AdvLatGAN-qua | AdvLatGAN-qua+ |
|---------|-----------|-------|--------|---------------|----------------|
| IS | DCGAN | $6.21 \pm 0.07$ | $6.03 \pm 0.03$ | $6.28 \pm 0.04$ | $\mathbf{6.58 \pm 0.34}$ |
|    | WGAN | $4.10 \pm 0.01$ | $6.84 \pm 0.04$ | $7.21 \pm 0.04$ | $\mathbf{7.76 \pm 0.07}$ |
|    | WGAN-GP | $6.80 \pm 0.02$ | $7.41 \pm 0.06$ | $7.60 \pm 0.06$ | $\mathbf{8.59 \pm 0.10}$ |
|    | SNGAN | $7.21 \pm 0.03$ | $6.30 \pm 0.06$ | $7.58 \pm 0.03$ | $\mathbf{8.13 \pm 0.06}$ |
| FID | DCGAN | $45.3 \pm 0.4$ | $41.9 \pm 0.4$ | $41.7 \pm 1.0$ | $\mathbf{40.1 \pm 1.5}$ |
|     | WGAN | $89.7 \pm 1.3$ | $1.3 \pm 0.4$ | $27.3 \pm 0.7$ | $\mathbf{27.2 \pm 0.5}$ |
|     | WGAN-GP | $32.3 \pm 0.3$ | $24.5 \pm 0.1$ | $22.6 \pm 0.4$ | $\mathbf{18.3 \pm 1.1}$ |
|     | SNGAN | $26.7 \pm 0.5$ | $50.9 \pm 2.7$ | $22.3 \pm 0.5$ | $\mathbf{21.9 \pm 0.3}$ |

### G.3 Experimental Comparison over Latent Exploration

We compare AdvLatGAN-z with representative works EvolGAN, DOT, DDLS that interplaying latent space exploration and GANs. The results are presented in Table. 3. For detailed experimental settings, please refer to Appendix H.3.

### G.4 Experimental Comparison over GAN Model with Adversarial Training

We compare AdvLatGAN-qua and AdvLatGAN-qua+ with two representative works ASGAN and Robust GAN training that combine adversarial training and GANs. These two methods are related to perturbations on the generated samples and the real samples respectively. The experimental setting is aligned with Table 4 and we evaluate on CIFAR-10 with four frameworks: DCGAN, WGAN, WGANGP and SNGAN. Inception Score and Fréchet Inception Distance are adopted as the evaluation metrics. The experimental results are presented in Table 12.

## H   Experimental Details for Post-training Latent Sampling: AdvLatGAN-z

### H.1 Details for Synthetic Data Experiment

**Protocols for Synthetic Experiment.** We simulate two synthetic datasets. Ring dataset is a mixture of 8 2-D Gaussians with mean $\{(2\cos(i\pi/4), 2\cos(i\pi/4))\}_{i=1}^{8}$ and standard deviation 0.001. 12.5K samples are simulated from each Gaussian distribution. 50K samples from $p(\mathbf{z})$ are used to generate $\mathbf{x}$ for test. Grid dataset is a mixture of 25 2-D isotropic Gaussians with mean $\{(2i, 2j)\}_{i,j=-2}^{2}$ and standard deviation 0.0025. 4K samples are simulated from each Gaussian. 20K samples from $p(\mathbf{z})$ are used to generate target samples $\{\tilde{\mathbf{x}}\}$ for test. In the synthetic experiments, we apply fully-connected networks for generation and the architectures are shown in Table 13 and Table 14.

<table>
<tr><td colspan="3">Table 13: Architecture of generator $G$.</td></tr>
<tr><td>Layer</td><td>Output size</td><td>Activation</td></tr>
<tr><td>Linear</td><td>100</td><td>ReLu</td></tr>
<tr><td>Linear</td><td>200</td><td>ReLu</td></tr>
<tr><td>Linear</td><td>100</td><td>ReLu</td></tr>
<tr><td>Linear</td><td>2</td><td>-</td></tr>
</table>

<table>
<tr><td colspan="3">Table 14: Architecture of discriminator $D$.</td></tr>
<tr><td>Layer</td><td>Output size</td><td>Activation</td></tr>
<tr><td>Linear</td><td>100</td><td>ReLu</td></tr>
<tr><td>Linear</td><td>200</td><td>ReLu</td></tr>
<tr><td>Linear</td><td>100</td><td>ReLu</td></tr>
<tr><td>Linear</td><td>1</td><td>-</td></tr>
</table>

**Details for Ring Data Experiment.** To better demonstrate the practical effectiveness, we insufficiently train the model with 2,000 iterations (1 epoch). The generated results by 20,000 samples in the 2D-Gaussian distribution of the insufficiently trained model are presented in Fig. 12 (a). AdvLatGAN-z is then applied to the latent samples under $\ell_\infty$ constraint. The step size is set as 0.003 and the transform is conducted by 8,000 iterations. The results of AdvLatGAN-z is presented in Fig. 12 (b) while Fig. 12 (c) presents the generated results of the sufficient trained model with 100 epochs. As the targeted real-data locates in 8 separate modes, samples from a continuous distribution via a continuous mapping cannot avoid invalid samples in the between of the targeted modes, even for the sufficiently trained model. While the latent distribution transform can well address the issue.

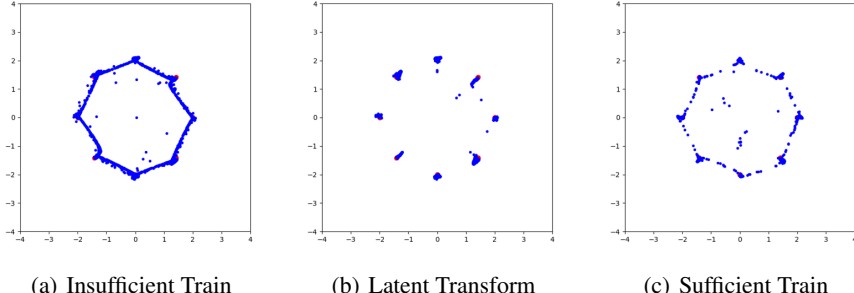

(a) Insufficient Train     (b) Latent Transform     (c) Sufficient Train

Figure 12: Left: generated results of insufficiently trained networks. Middle: generated results of AdvLatGAN-z transformed samples. Right: generated results of insufficiently trained networks.

.

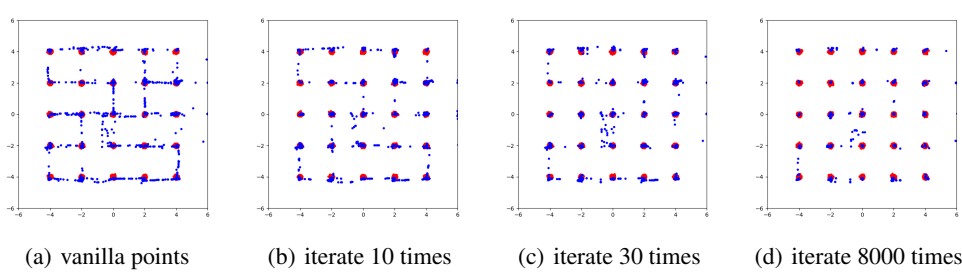

(a) vanilla points     (b) iterate 10 times     (c) iterate 30 times     (d) iterate 8000 times

Figure 13: Results of AdvLatGAN-z for sufficiently trained nets on Grid dataset.

.

**Details for Grid Data Experiment.** Complementary to the Ring experiment, in Grid setting, we show that AdvLatGAN-z is also effective for sufficiently trained networks. The generative networks are trained by 50 epochs while other hyper-parameters maintain consistent with Appendix H.1. The results are presented in Fig. 12, proving the effectiveness of AdvLatGAN-z in boosting generative performance.

## H.2 Details for MNIST Experiment in Fig.5

DCGAN and WGAN-GP are adopted as the backbones. The model structures are consistent with those used in Appendix J.1.1. AdvLatGAN-z is conducted with a total of 30 iterations, each with a step size of 0.03 and a batch size of 8000. The configuration of the trained DCGAN model: batch size is 128; D-learning rate and G-learning rate are both 0.0002; the loss function is BCE loss; the discriminator updates one step per generator iteration; the latent dimension is 2; and the total training step is 20,000. The configuration of training WGAN-GP model: the batch size is 128; the D-learning rate and G-learning rate are both 0.0002; the parameters of discriminator updates one step per generator iteration; the latent dimension is 2; and the total training step is 50,000.

## H.3 Details for STL-10 Experiment in Table 3

DCGAN, SNGAN and WGAN-GP are adopted as the backbones. The model structures are consistent with those used in Appendix J.1.1. AdvLatGAN-z is conducted for a total of 20 iterations, each with step size of 0.05 and the batch size is set as 10000.

The training configuration of DCGAN model: the batch size is 128; the D-learning rate and G-learning rate are both 0.0002; the loss function is bce loss; the discriminator updates one step per generator iteration; the latent dimension is 100; and the total training step is 100,000. The training configuration of SNGAN model: the loss function is hinge loss and other configurations are the same with DCGAN. The training configuration of the trained WGAN-GP model: the loss function is wasserstein loss and other configurations are the same with DCGAN. The visualization results of WGANGP for AdvLatGAN-z are presented in Fig. 14.

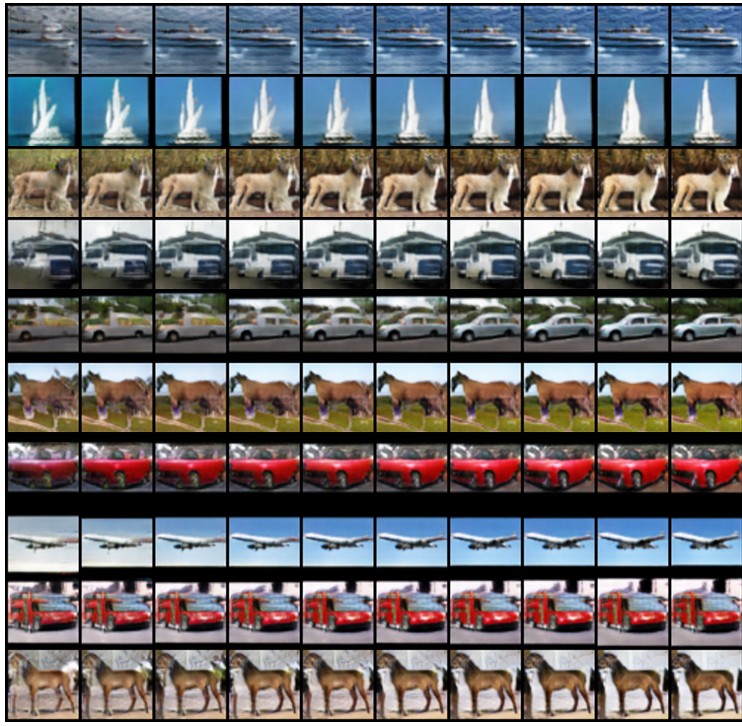

Figure 14: Visualization results of AdvLatGAN-z on STL-10. The left column shows the raw samples from the Gaussian, while samples from left to right present the generative results of the process of AdvLatGAN-z transform.

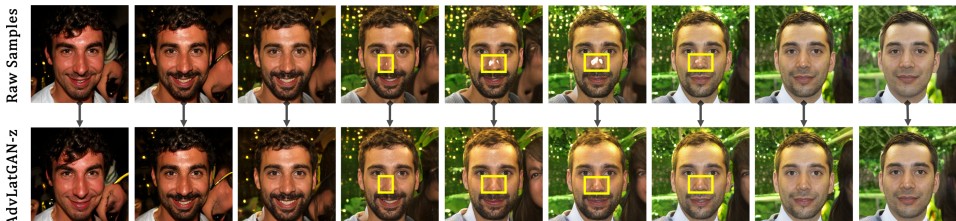

Figure 15: Results of AdvLatGAN-z on FFHQ dataset. First row: generated results of the interpolation of raw samples from Gaussian in Fig. 3. Second row: generated results under AdvLatGAN-z. AdvLatGAN-z can effectively mitigate defects.

### H.4 Details for AFHQ and FFDQ Experiment in Fig. 10

We adopt the pre-trained StyleGAN2-ada as the backbone referencing the official implementation. We initialize six latent space vectors from Gaussian which deliver bad generation and then perform AdvLatGAN-z to the original sampling. The generative results of the newly mined $\mathbf{z}s$ are presented in the second row. The step size of AdvLatGAN-z is set as 0.001 and the number of steps is set as 150. Taking the second row of Fig. 3 as example, we also apply AdvLatGAN-z to all steps in the interpolation as the supplementary results to Fig. 10. The results are presented in Fig. 15.

## I Experiment for Hard Sample Pair Mining for Diversity

The proposed AdvLatGAN-div involves mining the hard sample pair for diversity using $\mathbf{z}$ transform guided by Eq. 8 in GAN's bi-level optimization process. In Table. 1 and Fig. 6, we show the effectiveness of the proposed hard sample mining approach. Here we provide details for the conducted experiments and supplement the visualization results of CIFAR-10.

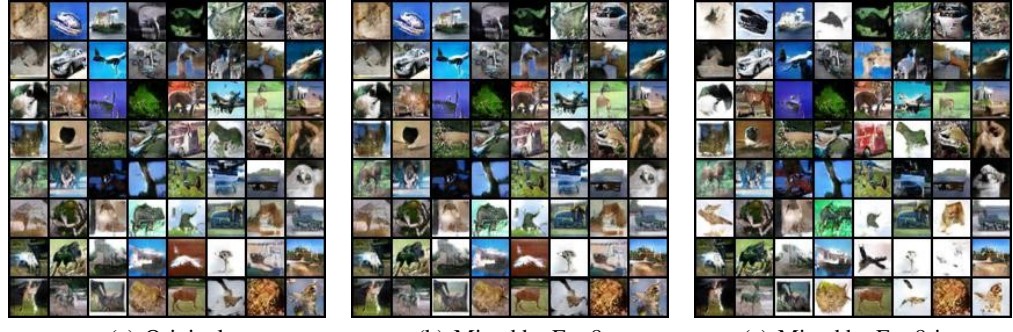

| (a) Original | (b) Mined by Eq. 8 | (c) Mined by Eq. 8-inv |

Figure 16: Results of diversity driven iterative transform scheme in latent space by using DCGAN [47] pre-trained on CIFAR-10. The pair obtained by solving Eq. 8 tend to collapse (as they are hard samples), while the opposite leads to better diversity by using the inverse form of Eq. 8.

Table 15: $\epsilon$ in AdvLatGAN-qua for CIFAR-10 and STL-10 datasets

| $\epsilon$ | DCGAN | WGAN | WGAN-GP | SNGAN | LSGAN | WGAN-div | ACGAN |
|---|---|---|---|---|---|---|---|
| CIFAR-10 | 0.03 | 0.01 | 0.015 | 0.04 | 0.01 | 0.03 | 0.03 |
| STL-10 | 0.04 | 0.02 | 0.015 | 0.05 | 0.03 | 0.03 | - |

**Details for CIFAR-10 Experiment.** For an initial random latent vector of batch size 64, we respectively conduct 300-step gradient sign descent and 300-step gradient sign ascent on the ratio of the distance between the original image and the newly obtained image to the distance of the original input vector and the newly obtained input vector (which denoted as Eq. 8). The results are presented in Fig. 16. The step size is set as 0.03. The model is a pre-trained CIFAR-10 DCGAN model.

**Details for FACADES Experiment.** Basically following the same pipeline as in Sec. I, we select six initial latent space vectors and utilize Eq. 8 and its inverse form to obtain the corresponding hard pair and easy pair. The hyper-parameters remain the same as Sec. I while the backbone is MSGAN trained on FACADES.

# J   Details of Experiment for Generative Map Improvement: AdvLatGAN-qua/div

## J.1   Experiment for AdvLatGAN-qua

To reduce the computational cost, for all the experiments in this section, we only take one I-FGSM step for updating the raw latent sampling during training.

### J.1.1   Details for CIFAR-10 and STL-10 datasets in Table 4

Recall that $\epsilon$ is the radius of the closed ball to $\ell_\infty$ constraint perturbation. For different compared baselines (backbones), training's $\epsilon$ settings are shown in Table 15. The training hyper-parameter configuration: the batch size is 128; the D-learning rate and G-learning rate are 0.0002; the latent dimension is 100; and the total training step is 100,000. In CIFAR-10 experiment, the basic DCGAN architectures for $G$ and $D$ are presented in Table 17 and Table 18. While in STL-10 experiment, the architectures are presented in Table 19 and Table 20. Evaluation metrics are calculated over 50000 images. The iteration step size $\epsilon$ of AdvLatGAN-z is set to 0.05 and we conduct 20 steps each time.

Table 16: $\epsilon$ in AdvLatGAN-qua for larger scale datasets

| $\epsilon$ | LSUN-64 | LSUN-128 | CelebA-64 | CelebA-128 | ImageNet |
|---|---|---|---|---|---|
| SNGAN | 0.05 | 0.05 | 0.05 | 0.006 | 0.05 |
| WGAN-GP | 0.015 | 0.015 | 0.02 | 0.015 | 0.03 |

Table 17: Architecture of generator $G$ in CI-FAR10 experiment.

| Layer | Output size |
|---|---|
| ConvTranspose2d | $2 \times 2 \times 1024$ |
| BatchNorm2d | $2 \times 2 \times 1024$ |
| Relu | $2 \times 2 \times 1024$ |
| ConvTranspose2d | $4 \times 4 \times 512$ |
| BatchNorm2d | $4 \times 4 \times 512$ |
| Relu | $4 \times 4 \times 512$ |
| ConvTranspose2d | $8 \times 8 \times 256$ |
| BatchNorm2d | $8 \times 8 \times 256$ |
| Relu | $8 \times 8 \times 256$ |
| ConvTranspose2d | $16 \times 16 \times 128$ |
| BatchNorm2d | $16 \times 16 \times 128$ |
| Relu | $16 \times 16 \times 128$ |
| ConvTranspose2d | $32 \times 32 \times 3$ |
| Tanh | $32 \times 32 \times 3$ |

Table 18: Architecture of discriminator $D$ in CIFAR10 experiment.

| Layer | Output size |
|---|---|
| Conv2d | $16 \times 16 \times 64$ |
| LeakyRelu | $16 \times 16 \times 64$ |
| Conv2d | $8 \times 8 \times 128$ |
| LeakyRelu | $8 \times 8 \times 128$ |
| BatchNorm2d | $8 \times 8 \times 128$ |
| Conv2d | $4 \times 4 \times 256$ |
| LeakyRelu | $4 \times 4 \times 256$ |
| BatchNorm2d | $4 \times 4 \times 256$ |
| Conv2d | $2 \times 2 \times 512$ |
| LeakyRelu | $2 \times 2 \times 512$ |
| BatchNorm2d | $2 \times 2 \times 512$ |
| faltten | 2048 |
| linear | 1 |

### J.1.2 Details for larger scale datasets in Table 5

We evaluate AdvLatGAN-qua and AdvLatGAN-qua+ on larger scale data including LSUN Church, CelebA and ImageNet datasets with image resolution up to $64 \times 64$ and $128 \times 128$. We adopt WGANGP and SNGAN as the baselines (backbones) with basically the same training hyper-parameter configuration to Appendix J.1.1 except for the number of trained iterations. We train LSUN-64 for 100,000 iterations, LSUN-128 for 150,000 iterations, CelebA-64 for 100,000 iterations, CelebA-128 for 300,000 iterations and ImageNet-128 for 500,000 iterations. For image resolution of $64 \times 64$, SNGAN's architectures for $G$ and $D$ are presented in Table. 21 and Table.23. While for $128 \times 128$, the architectures follow Table. 22 and Table.24. Training's $\epsilon$ settings are shown in Table 16. FID is calculated over 50000 images while density and coverage are calculated over 10000 images. The iteration step size $\epsilon$ of AdvLatGAN-z is set as 0.01 for SNGAN and 0.002 for WGANGP and we conduct 100 steps each time. The visualization results are presented in Fig. 17, Fig. 18, Fig. 19, Fig. 20 and Fig. 21.

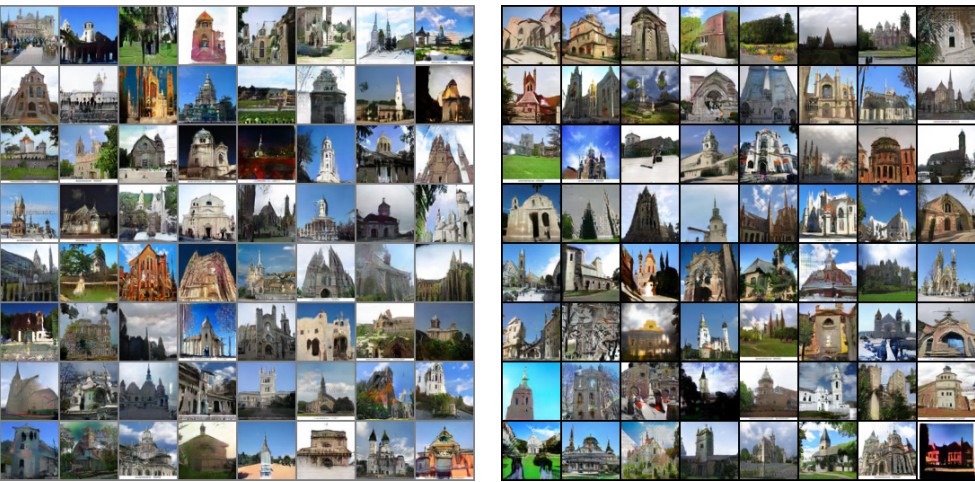

(a) Original                    (b) AdvLatGAN-qua+

Figure 17: Generative results on LSUN-64.

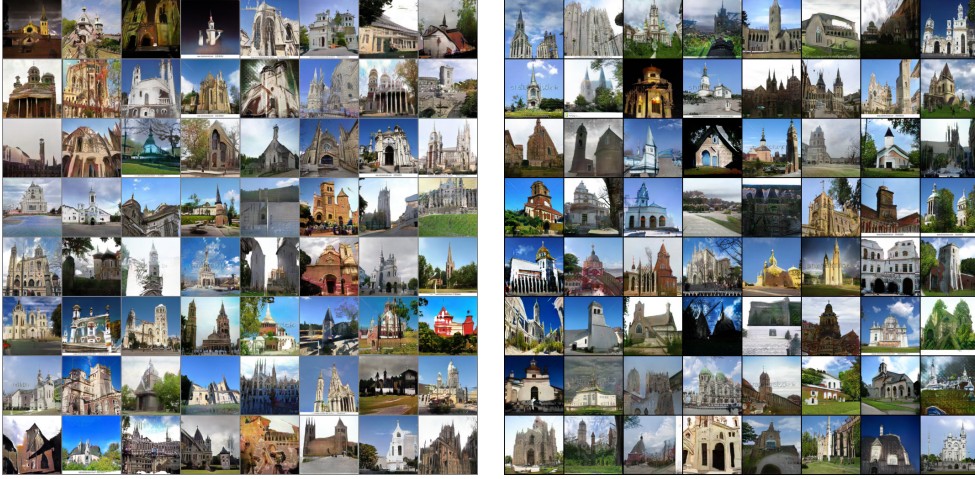

(a) Original                    (b) AdvLatGAN-qua+

Figure 18: Generative results on LSUN-128.

Table 19: Architecture of generator $G$ in STL-10 experiment.

| Layer | Output size |
|---|---|
| ConvTranspose2d | $3 \times 3 \times 1024$ |
| BatchNorm2d | $3 \times 3 \times 1024$ |
| Relu | $3 \times 3 \times 1024$ |
| ConvTranspose2d | $6 \times 6 \times 512$ |
| BatchNorm2d | $6 \times 6 \times 512$ |
| Relu | $6 \times 6 \times 512$ |
| ConvTranspose2d | $12 \times 12 \times 256$ |
| BatchNorm2d | $12 \times 12 \times 256$ |
| Relu | $12 \times 12 \times 256$ |
| ConvTranspose2d | $24 \times 24 \times 128$ |
| BatchNorm2d | $24 \times 24 \times 128$ |
| Relu | $24 \times 24 \times 128$ |
| ConvTranspose2d | $48 \times 48 \times 3$ |
| Tanh | $48 \times 48 \times 3$ |

Table 20: Architecture of discriminator $D$ in STL-10 experiment.

| Layer | Output size |
|---|---|
| Conv2d | $24 \times 24 \times 64$ |
| LeakyRelu | $24 \times 24 \times 64$ |
| Conv2d | $12 \times 12 \times 128$ |
| LeakyRelu | $12 \times 12 \times 128$ |
| BatchNorm2d | $12 \times 12 \times 128$ |
| Conv2d | $6 \times 6 \times 256$ |
| LeakyRelu | $6 \times 6 \times 256$ |
| BatchNorm2d | $6 \times 6 \times 256$ |
| Conv2d | $3 \times 3 \times 512$ |
| LeakyRelu | $3 \times 3 \times 512$ |
| BatchNorm2d | $3 \times 3 \times 512$ |
| faltten | 4608 |
| linear | 1 |

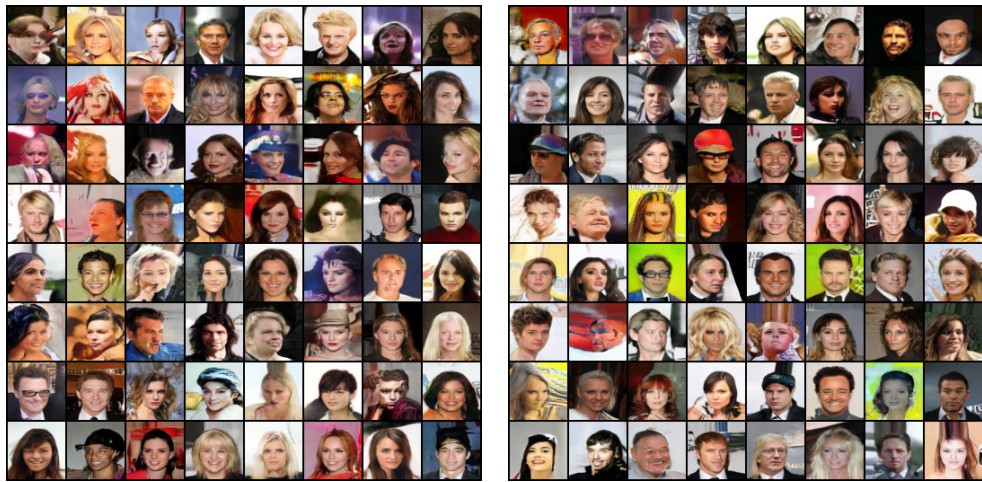

(a) Original             (b) AdvLatGAN-qua+

Figure 19: Generative results on CelebA-64.

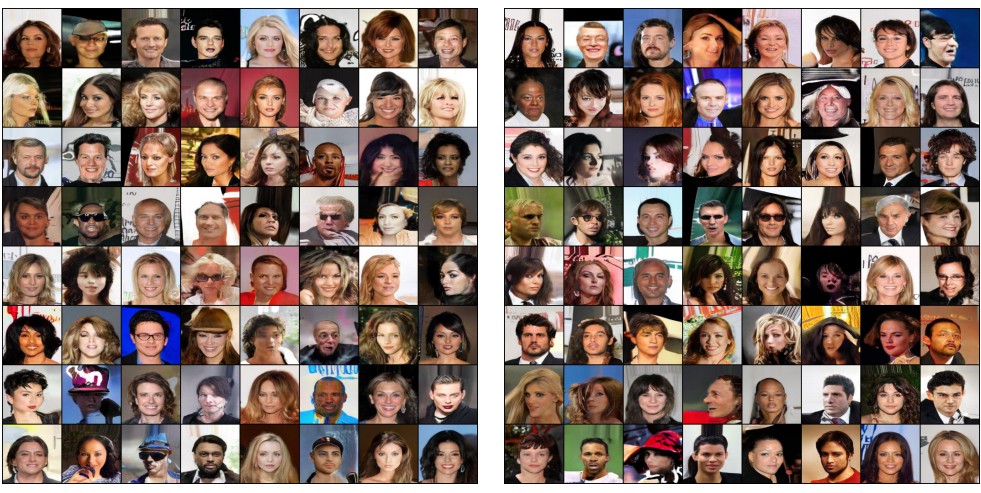

(a) Original             (b) AdvLatGAN-qua+

Figure 20: Generative results on CelebA-128.

Table 21: Architecture of generator $G$ for SNGAN-64

| Layer | Output size |
| --- | --- |
| Linear+Reshape | $8 \times 8 \times 512$ |
| ConvTranspose2d | $16 \times 16 \times 256$ |
| BatchNorm2d | $16 \times 16 \times 256$ |
| Relu | $16 \times 16 \times 256$ |
| ConvTranspose2d | $32 \times 32 \times 128$ |
| BatchNorm2d | $32 \times 32 \times 128$ |
| Relu | $32 \times 32 \times 128$ |
| ConvTranspose2d | $64 \times 64 \times 64$ |
| BatchNorm2d | $64 \times 64 \times 64$ |
| Relu | $64 \times 64 \times 64$ |
| Conv2d | $64 \times 64 \times 3$ |
| Tanh | $64 \times 64 \times 3$ |

Table 22: Architecture of generator $G$ for SNGAN-128

| Layer | Output size |
| --- | --- |
| Linear+Reshape | $16 \times 16 \times 512$ |
| ConvTranspose2d | $32 \times 32 \times 256$ |
| BatchNorm2d | $32 \times 32 \times 256$ |
| Relu | $32 \times 32 \times 256$ |
| ConvTranspose2d | $64 \times 64 \times 128$ |
| BatchNorm2d | $64 \times 64 \times 128$ |
| Relu | $64 \times 64 \times 128$ |
| ConvTranspose2d | $128 \times 128 \times 64$ |
| BatchNorm2d | $128 \times 128 \times 64$ |
| Relu | $128 \times 128 \times 64$ |
| Conv2d | $128 \times 128 \times 3$ |
| Tanh | $128 \times 128 \times 3$ |

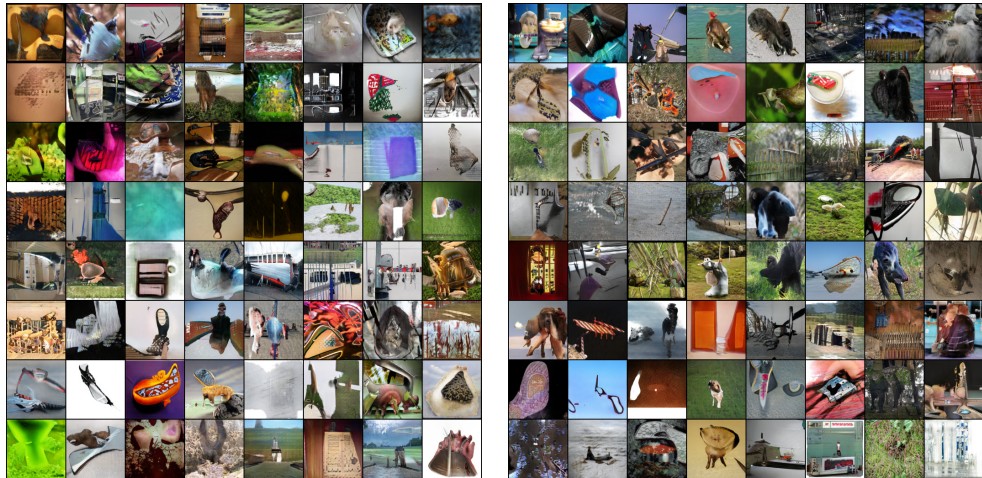

(a) Original                                         (b) AdvLatGAN-qua+

Figure 21: Generative results on ImageNet-128.

Table 23: Architecture of discriminator $D$ for SNGAN-64

| Layer | Output size |
|---|---|
| SN+Conv2d | $64 \times 64 \times 64$ |
| LeakyReLU | $64 \times 64 \times 64$ |
| SN+Conv2d | $32 \times 32 \times 64$ |
| LeakyReLU | $32 \times 32 \times 64$ |
| SN+Conv2d | $32 \times 32 \times 128$ |
| LeakyReLU | $32 \times 32 \times 128$ |
| SN+Conv2d | $16 \times 16 \times 128$ |
| LeakyReLU | $16 \times 16 \times 128$ |
| SN+Conv2d | $16 \times 16 \times 256$ |
| LeakyReLU | $16 \times 16 \times 256$ |
| SN+Conv2d | $8 \times 8 \times 256$ |
| LeakyReLU | $8 \times 8 \times 256$ |
| SN+Conv2d | $8 \times 8 \times 512$ |
| LeakyReLU | $8 \times 8 \times 512$ |
| Faltten+Linear | 1 |

Table 24: Architecture of discriminator $D$ for SNGAN-128

| Layer | Output size |
|---|---|
| SN+Conv2d | $128 \times 128 \times 64$ |
| LeakyReLU | $128 \times 128 \times 64$ |
| SN+Conv2d | $64 \times 64 \times 64$ |
| LeakyReLU | $64 \times 64 \times 64$ |
| SN+Conv2d | $64 \times 64 \times 128$ |
| LeakyReLU | $64 \times 64 \times 128$ |
| SN+Conv2d | $32 \times 32 \times 128$ |
| LeakyReLU | $32 \times 32 \times 128$ |
| SN+Conv2d | $32 \times 32 \times 256$ |
| LeakyReLU | $32 \times 32 \times 256$ |
| SN+Conv2d | $16 \times 16 \times 256$ |
| LeakyReLU | $16 \times 16 \times 256$ |
| SN+Conv2d | $16 \times 16 \times 512$ |
| LeakyReLU | $16 \times 16 \times 512$ |
| Faltten+Linear | 1 |

### J.1.3   Additional Experiments on StyleGAN2-ada Backbone.

We evaluate AdvLatGAN-qua on StyleGAN2-ada [18] backbone on MetFaces [18] and AFHQ Cat datasets. The implementation is based on the official PyTorch implementation of [18]. We adopt FID and kernel inception distance (KID) [64] as evaluation metrics in line with [18]. All the training settings are inline with the default setting of [18]'s official code. We report results of the best generation (referring to KID). The quantitative results show the significant performance gain.

### J.2   Experiment for AdvLatGAN-div

### J.2.1   Details for CIFAR-10 Expriment

We adopt three latent vector updates per generator iteration and the step size is set as 0.01. Recall that the transform for mining samples starts at the Gaussian neighborhood of the initial latent vector, here the standard deviation of the Gaussian is set as 0.01. Other training parameters are referred to the original MSGAN implementation. FID, density and coverage metrics for each class are calculated

Table 25: Evaluation of AdvLatGAN-qua on StyleGAN2-ada backbone.

| Dataset | StyleGAN2-ada | | AdvLatGAN-qua | |
|---|---|---|---|---|
| | FID($\downarrow$) | KID$\times 10^3$($\downarrow$) | FID($\downarrow$) | KID$\times 10^3$($\downarrow$) |
| AFHQ Cat-128 | 4.516 | 0.906 | **3.742** | **0.771** |
| AFHQ Cat-512 | 4.133 | 0.940 | **3.224** | **0.749** |
| MetFaces-128 | 22.328 | 6.159 | **20.952** | **4.524** |
| MetFaces-1024 | 19.420 | 3.132 | **18.698** | **2.697** |

based on 5000 images, and the overall evaluation is performed on the collection of generated images of all classes (50000 in total). The iteration step size $\epsilon$ of AdvLatGAN-z is set to 0.01 and we conduct 100 steps each time.

### J.2.2 Details for STL-10 Expriment

The basic experimental setup in STL-10 experiment follows Appendix J.2.1, whereas the difference is that: for STL-10, we use the training part of the dataset as the real data for training, while for evaluation, the metrics are calculated using the test part of the dataset.