# OpenReview forum: "Improving Generative Adversarial Networks via Adversarial Learning in Latent Space"
_NeurIPS.cc/2022/Conference — NeurIPS 2022 Accept_

### Official Review · Reviewer_phhm · 2022-07-06

**Rating:** 5
**Confidence:** 4
**Soundness:** 2 fair
**Presentation:** 2 fair
**Contribution:** 2 fair

**Summary:**

Author proposed a novel idea about training/sampling GAN from a discontinuous distribution by setting one more layer in-between known prior $P_z$ and real distribution $P_x$. Author provided mathematical proof that discontinuity of latent space (support of G) is required to cover the complex and discontinuous real distribution. It is also provided that minimizing Eq.5 is equivalent to minimizing Jenson Shanon Divergence between the latent distribution $P_z$ and a distribution of a union of disconnected subset in the latent space $P_z^{op}$. The effectiveness of the proposed idea is shown in Experiments.

**Questions:**

1. It is not understandable that if real distribution is a union of disconnected manifolds, the support of G needs to be disconnected as well.

Line 534-536: Let's say $U_1$ is Male and $U_2$ is Female. The proof goes like this; assume there exists a latent vector $z$ generating both Female and Male, i.e., $G(z) \in U_1$ and $G(z) \in U_2$. This implies $z \in U_1 \cap U_2$, which contradicts the condition 'Female and Male do not have intersection', i.e., $U_1 \cap U_2 = \emptyset$. To me, it looks like a simple repeat of the assumption and does not prove that $Z^{op}(G)$ is a union of disconnected subsets in the latent space. Please correct me If I am wrong.

And If my argument is not wrong, the entire logical flow of this paper must be revised as well.

2. The proof in Proposition B.3. includes an assumption that $E_{x \sim p_r}[log(D(x))]=E_{z \sim p_z^{op}}[log(D(G(z)))]$, which means the given generator (hence the discriminator as well) for the proof must be optimal. It needs to be mentioned. Furthermore, in practice in Alg. 1, optimizing Eq. 5 is conducted without optimal $G$ and $D$, which needs to be at least justified in the paper. Please correct me if I am wrong.

3. I am not sure, but isn't it bad in terms of diversity if $z$ is optimized by Eq.5? What is the number of real/fake samples for FID score in Table 4?

**Limitations:**

Limitation is not provided in the paper.

**Strengths And Weaknesses:**

$\textbf{Strengths}$
1. Proposed idea is interesting and may have a  possibility to be used to avoid from unrealistic generation from GAN.
2. Experiment result verifies the effectiveness of the proposed idea.

$\textbf{Weakness}$
1. Logical flow of the paper at Introduction part and connection between the motivation of the paper and the proposed method is not clear.
Specifically, the logical flow is like below:

  (1.1). Real distribution is a union of disconnected manifolds

  (1.2). Proof for necessity of disconnected latent space (at least in high level)

  (1.3). The gaussian space of ${G}$ is not disconnected.

  (1.4). Eq. 5 and Eq 8?

Here, (1.2) is not mentioned in Introduction, which I think necessary.
More importantly, (1.4) has a weak connection with the logical flow.

2. Mathematical proof: I wrote my thought in the Questions part.

3. Method part is written well, but Introduction part is NOT organized well. It also has some unnecessary or confusing expressions.

For example,

at Line 50, “otherwise the generative expression will differ.” What does the generative expression stand for? Why is it necessary for justifying the perturbation-based method?

at Line 30-31, "One reason is that the generator is a continuous function (as neural nets) while the generation quality (e.g. natural images in pixel space) is not." Does it mean the generation quality is not a continuous function? How can the quality be a function?

at Line 38-40, Most generative models such as GAN, VAE, Diffusion Model and Flow-based model also inherently have a mapping between simple known distribution and complex real distribution. I believe disconnected modes of real distribution cannot be said to be "well-known difficulty in GANs training".

---

> ### Author Response · Authors · 2022-08-02
> **Response to Reviewer phhm**
>
> Thanks for your time and valuable comments. We are pleased with your acknowledgment of our idea and experiments. Below we respond to your comments/questions point-by-point, reclaim our logical flow, revisit the mathematical proof, clarify unclear/confusing statements and have revised the paper accordingly.
>
> **Q1: Logical flow in Introduction section is not well organized and the connection between the motivation and the proposed method is not clear.**
>
> **R1:** Sorry for the confusion. Please refer to the general response for our summarized logical flow noted with the corresponding paragraph in the paper. Your logical flow is more likely to be the sub logical flow of our (1) and (4). We have reorganized our Introduction Section and rephrased the statements that cause confusion in our current paper. The connection between the motivation and the proposed method is also included in our logical overview in the general response and we have emphasized such connection for AdvLatGAN-z (as you seem to care more about this logical line reflected by your logical flow) in Line 51-54 and Line 170-173.
>
> **Q2: Concerns about Proposition 3.2.**
>
> **R2:** We sincerely appreciate your careful verification of the proof. However, it seems that you might not clearly distinguish between latent space and pixel space or do not take $G$ as a given function with fixed parameters (we have added the word "given" to emphasize it in Proposition 3.2). Following your notations, please note $U_1$ and $U_2$ are in the codomain of $G$ (in pixel space) while $\mathbf{z}$ is in the domain of $G$ (in latent space). Thus "$z \in U_1 \cap U_2$" in your logic is mathematically wrong (it should be "$G(z) \in U_1 \cap U_2$"). Line 544-546 tend to illustrate if $U_1 \cap U_2 = \emptyset$ then $G^{-1}(U_1) \cap G^{-1}(U_2)=\emptyset$. The proof goes like: if $G^{-1}(U_1) \cap G^{-1}(U_2)\neq\emptyset$, then there  exists a latent vector $\mathbf{z}$ such that $z\in G^{-1}(U_1)$ and $z \in G^{-1}(U_2)$. This implies $\mathbf{z}$ can generate both $U_1$ and $U_2$ and $G(z) \in U_1 \cap U_2$, which contradicts the condition $U_1 \cap U_2 = \emptyset$. Please note that Proposition 3.2 is acknowledged by Review qXs3 as intuitive and makes sense. Please let us know if you have further questions.
>
>
> **Q3: The proof in Proposition B.3. includes an assumption that $E_{x\sim p_r}[log(D(x))]=E_{z\sim p_z^{op}}[log(D(G(z)))]$, which means the given generator (hence the discriminator as well) for the proof must be optimal.**
>
> **R3:** The proof does require $E_{x\sim p_r}[log(D(x))]=E_{z\sim p_z^{op}}[log(D(G(z)))]$ to be satisfied, but please note that this equation is guaranteed by the definition of $p_z^{op}(G)$ (Definition 3.3) that $p^{op}_z(G)$ satisfies if $z \sim p^{op}_z(G)$ then $G(z) \sim p_r$. For any given generator $G$, there will be a corresponding $p^{op}_z(G)$, and it does not require the generator to be optimal. Likewise, this equation will be satisfied for any given $D$, thus there are also no constraints for the discriminator.
>
> **Q4: Isn't it bad in terms of diversity if $\mathbf{z}$ is optimized by Eq.5.**
>
> **R4:** Thanks for the valuable question. Eq. 5 may indeed cause a slight diversity drop compared to the generation results of raw Gaussian samples. However, please note that this diversity deviation is brought by the bad generations of raw Gaussian samples, which is meaningless for realistic generation. For example, natural images mixed with bad generation results will have a better diversity than pure natural images, as bad generations have a more different distribution from natural images. On the other hand, as we bound the perturbation of $\mathbf{z}$, the diversity drop will only be slight, while the quality gain can be significant.
>
> **Q5: What is the number of real/fake samples for FID score in Table 4?**
>
> **R5:** Evaluation metrics are calculated over 50k real samples and 50k fake samples. The experimental details are in the appendix. Please refer to Line 731-732.
>
> **Q6: Line 30-31: Does it mean the generation quality is not a continuous function? How can the quality be a function?**
>
> **R6:**  Thanks for noting. We have rephrased it: "... the generation quality reflected by the matching degree to the natural image distribution in pixel space does not exhibit a continuous nature as two nearby images' quality can differ much."

---

> > ### Author Response · Authors · 2022-08-02
> > **Response to Reviewer phhm (Cont.)**
> >
> > **Q7: Line 52: “otherwise the generative expression will differ.” What does the generative expression stand for? Why is it necessary for justifying the perturbation-based method?**
> >
> > **R7:** The generative expression stands for the main content of the generated image. We have rephrased the sentences to fix the ambiguity and emphasize why we choose the perturbation-based method: "We model the $\mathbf{z}$ transform process as making perturbations to the original sampling since $z^*(\mathbf{z})$ shall not depart much from $\mathbf{z}$ as we hope the main content of the generated image remains the same. The hope that small perturbations can achieve considerable positive quality variation leads us to the adversarial sample mining methods." There can be other designs of $z^*$, but we believe adversarial sample mining methods are one of the effective solutions (we have also tried raw gradient updating, but it performs much worse), and note that qualitative/quantitative results in Table 4 and Figure 4 have verified its effectiveness.
> >
> >
> > **Q8: Line 40-42: Most generative models such as GAN, VAE, Diffusion Model and Flow-based model also inherently have a mapping between simple known distribution and complex real distribution. I believe disconnected modes of real distribution cannot be said to be "well-known difficulty in GANs training".**
> >
> > **R8:** Sorry for the ambiguity. We mean that the continuity contradiction can be one important factor that leads to the well-known training difficulty of GANs i.e. GANs are often unstable to train and discriminators are more easily optimized than generators. We have rephrased it: "... can be one of the significant factors leading to the well-known training difficulty/instability of GANs."
> >
> > ---
> > We hope this response could help address your concerns. We believe our work can contribute to the research in GAN with methodological novelty and strong empirical results. We would sincerely appreciate it if you could reconsider your rating and wish to receive your further feedback soon.

---

> > > ### Comment · Reviewer_phhm · 2022-08-10
> > > **Response**
> > >
> > > Most of my concerns have been settled during the rebuttal.
> > > I changed my score to 5.
> > > Thanks for the clarification.
> > >
> > > Additionally, I believe it would be better if the "evaluation metric descriptions" are in main paper rather than in Appendix since it is not trivial.

---

> > > > ### Author Response · Authors · 2022-08-10
> > > > **Thanks**
> > > >
> > > > Thanks for your nice feedback and valuable suggestion. Since we can no longer update the pdf, we will move the "evaluation metric descriptions" from Appendix to the main paper in our final version.

---

> ### Author Response · Authors · 2022-08-09
> **To Reviewer phhm: Looking Forward to Your Feedback**
>
> We are still looking forward to hearing your possible feedback. It is really a good chance for us to engage in the discussion to help us improve the paper.

---

### Official Review · Reviewer_Y24j · 2022-07-11

**Rating:** 8
**Confidence:** 4
**Soundness:** 4 excellent
**Presentation:** 3 good
**Contribution:** 3 good

**Summary:**

By introducing the adversarial techniques into GAN, this paper proposes new training and latent variable sampling methods to achieve better generation quality and diversity. The motivation is interesting that, the generator's limit as a continuous mapping: a) the contradiction between continuous mapping pixel-space and quality discontinuity; b) the mapping can incur mode collapse. Accordingly, the work adds a latent space sampling shift to amend the raw latent distribution and introduces targeted sampling shifts into GAN’s bi-level optimization for quality and diversity improvement. Extensive experiments on image data including large-scale dataset and backbone networks show that the proposed techniques lead to promising improvements.

**Questions:**

1.I can understand that the paper’s contribution is irrelevant to the specific choice of the backbone, and the experiments have already shown its effectiveness on various GAN backbones. Yet it is still informative to see its role on more SOTA designs e.g. StyleGAN.

2.I also understand that there exists a difference between qua (quality) and div (diversity) settings, but there is still room for synergizing the two parts. Why not try adopting qua and div at the same time? Or the authors shall better clarify this point.

3.Since many generative models other than GAN also use a single network to accomplish the generative task, do they also suffer from the same quality discontinuity issue? Can the proposed method be generalized to other generative models e.g. VAE?


**Limitations:**

Please see above.

**Strengths And Weaknesses:**

### Strengths:

1.This work is highly motivated and well presented. The idea is interesting.

2.The contributions are clear. This paper notes the effectiveness and importance of latent space sampling shift in GAN and may incur more follow-up studies devoted to making twofold efforts including sampling shift and mapping improvement.

3.The idea is supported by extensive evaluation with impressive results and theoretical arguments.

### Weaknesses:

1.The proposed latent sampling shift is iteratively conducted, which may incur more computational costs, especially in in-training algorithms. The trade-off between time consumption and performance gain needs to be discussed.

2.Some related works shall be discussed. The following papers mine latent space for different targets like conditional generation are worth for discussion.

[1] Nguyen A, Clune J, Bengio Y, et al. Plug & play generative networks: Conditional iterative generation of images in latent space, CVPR 2017.

[2] Nguyen A, Dosovitskiy A, Yosinski J, et al. Synthesizing the preferred inputs for neurons in neural networks via deep generator networks, NIPS 2016.

---

> ### Author Response · Authors · 2022-08-02
> **Response to Reviewer Y24j**
>
> Thanks for your time and valuable comments. We are encouraged that you appreciated our idea, contributions, experimental evaluation and theoretical support. Below we respond to your concerns and supplement new experiments as suggested to further strengthen our contributions.
>
> **Q1: The discussion of overhead.**
>
> **R1:** Please refer to Appendix D for the overhead experiments. The in-training algorithms do incur additional computational costs, but as we select a very small number of updating steps (1 for AdvLatGAN-qua and 3 for AdvLatGAN-div) in training, the additional overhead is small (around 20%-30%) while the performance gain is still significant. Compared to GAN's bi-level optimization process, the sampling method AdvLatGAN-z costs little, and it also outperforms peer latent sampling improvement methods in terms of cost-effectiveness in Table 8.
>
> **Q2: Some related works shall be discussed.**
>
> R2: Thanks for showing these works. These works [1] [2] explore another interesting application of latent variable updating to achieve conditional generation through an unconditional generative net, which generally perform gradient ascent in latent space to maximize the neuron activations in a separate classifier network. [1] extends [2] by introducing an additional prior on the latent code to further improve the generation. Since these efforts are devoted to a generally different sub-task i.e. achieving conditional generation through an unconditional model, the similarity may merely lie in that we all operate on the latent variables. We have added them in the related work section.
>
> **Q3: Suggestions to supplement experiments on StyleGAN backbone.**
>
> **R3:** Thanks for the suggestion. We evaluate AdvLatGAN-qua on StyleGAN2-ada [3] on 128-sized and full-sized MetFaces and AFHQ Cat datasets, and we are glad to report our positive results. The implementation is based on the official PyTorch implementation of [3]. We adopt Fréchet inception distance (FID) [4] and kernel inception distance (KID) [5] as evaluation metrics in line with [3]. All the training settings are in line with the default setting of [3]'s official code. We report the results of the best generation (referring to KID). We can see that on all four datasets, our method achieves significant performance gain through FID and KID metrics (best FID improvement from 4.133 to 3.224 on AFHQ Cat-512 and best KID improvement from 6.159 to 4.524). The results have been added and discussed in Appendix K.1.3.
>
> FID results (the lower the better):
>
> |               | AFHQ Cat-128 | MetFaces-128 | AFHQ Cat-512 | MetFaces-1024 |
> | ------------- | ------------ | ------------ | ------------ | ------------- |
> | StyleGAN2-ada | 4.516        | 22.328       | 4.133        | 19.420        |
> | AdvLatGAN-qua | **3.742**        | **20.952**       | **3.224**        | **18.698**        |
>
>
> KID$\scriptsize\times10^{3}$ results (the lower the better):
>
> |               | AFHQ Cat-128 | MetFaces-128 | AFHQ Cat-512 | MetFaces-1024 |
> | ------------- | ------------ | ------------ | ------------ | ------------- |
> | StyleGAN2-ada | 0.906        | 6.159        | 0.940         | 3.132          |
> | AdvLatGAN-qua | **0.771**        | **4.524**        | **0.749**        | **2.697**          |
>
>
> **Q4: Why not try adopting qua and div at the same time?**
>
> **R4:** We indeed had already tried with different weights to promote the combination, but the improvement is minor and unstable. Note there hardly exists successful efforts in explicitly modeling and unifying these two aspects. The difficulty lies in the contradiction that whether to cover more modes or to generate high-fidelity samples for generating a single image, which has already been discussed in the main paper (Line 33-38). We leave this nontrivial task for future work.
>
> **Q5: Do other generative models also suffer from the same quality discontinuity issue? Can the proposed method be generalized to other models?**
>
> **R5:** As long as the generative model is single neural network based which makes it a continuous mapping, it suffers from this issue. Because the generated results from a continuous latent distribution can not fully match the real distribution which lies in many disconnected manifolds. Please refer to Section 1 and 3.2 for detailed illustration.
>
> The proposed sampling method is specialized for GAN as its discriminator is a key element in the loss function, thus the method can not directly be adapted to other generative models. While our work may inspire future work on adapting our method or idea to other generative models beyond GAN.
>
> ---
> We hope this response could help address your concerns, and wish to receive your further feedback soon.

---

> > ### Author Response · Authors · 2022-08-02
> > **References**
> >
> > [1] Plug & play generative networks: Conditional iterative generation of images in latent space. CVPR 2017.
> >
> > [2] Synthesizing the preferred inputs for neurons in neural networks via deep generator networks. NIPS 2016.
> >
> > [3] Training generative adversarial networks with limited data. NeurIPS 2020.
> >
> > [4] GANs trained by a two time-scale update rule converge to a local Nash equilibrium. NeurIPS 2017.
> >
> > [5] Demystifying MMD GANs. ICLR 2018.

---

> ### Comment · Reviewer_Y24j · 2022-08-10
> **Reply to the author's response**
>
> I have read the reply from the author and other reviews. I appreciate the author's effort, all my concerns have been resolved. I think this paper is techenically sound. I tend to accept this paper and increase my score to 8.

---

### Official Review · Reviewer_Z7p2 · 2022-07-12

**Rating:** 5
**Confidence:** 4
**Soundness:** 3 good
**Presentation:** 4 excellent
**Contribution:** 3 good

**Summary:**

This paper studies the latent space in Generative Adversarial Nets (GANs) by introducing a implicit mapping on latent variable z before sending it to the generator G. Such a network z*(z) is trained by minimizing the generator loss, with G and D fixed. An additional level of optimization is introduced to optimize z* during the training of the proposed AdvLatGAN. Based on improving the quality and diversity of the model, two variants of the model are introduced, by modifying the objective of z*.

**Questions:**

Please see the previous session

**Limitations:**

It would be great to see the performance of AdvLatGAN on some larger-scale image dataset.

**Strengths And Weaknesses:**

Strength:

1) The paper presents an interesting perspective of improving GANs. The idea seems to be novel.
2) Experiments show improvements on Cifar-10 and STL image generation, by addion AdvLat in different GAN framework.

Weakness:

1) The idea of using implicit distribution for generative models has been explored in a few works, including [1], it would be great to have some comparisons with these methods.
2) The idea of adding additional network to transform z seems to be something could have been done within the generator, if we think of the transform network and the generator network as a whole. I'm wondering if the improvement was actually achieved by the bi-level optimization. It would be great to see some ablation study on the effectiveness of each component of the proposed method.


[1] Fang, L., Li, C., Gao, J., Dong, W., & Chen, C. (2019). Implicit deep latent variable models for text generation. arXiv preprint arXiv:1908.11527.

---

> ### Author Response · Authors · 2022-08-02
> **Response to Reviewer Z7p2**
>
> Thanks for your time and valuable comments. We are glad that you acknowledged our idea novelty and the experimental evaluation. Below we respond to your specific comments.
>
> **Q1: Discussion about implicit distribution related generative models.**
>
> **R1:** Thanks for sharing the work. For the mentioned work [1], it improves VAE by replacing the Gaussian variational posterior distribution constraining the encoded $\mathbf{z}$ with a sample-based distribution, which is less constrained and more expressive. Though interesting, this work may be of little relevance to us. [1] is specialized for VAE and the "implicit distribution" it improves refers to the variational posterior which is generally an exclusive component of VAE, while the prior Gaussian distribution, which is a more similar concept to our latent distribution of GAN, remains unchanged in [1] compared to the vanilla VAE. This work focuses more on the improvements in VAE design rather than the continuity concern or more efficient sampling in the prior latent distribution. On the other hand, [1] works on the text generation task via sequential model which also differs from ours.
>
> In fact, we have already discussed many latent distribution sampling improvement methods (Line 93-102) which potentially modify the sampling distribution in latent space to an implicit one, including a quantitative experimental comparison in Table 3.
>
> [1] Implicit deep latent variable models for text generation. arXiv preprint arXiv:1908.11527.
>
> **Q2: The idea of adding an additional network to transform z seems to be something could have been done within the generator.**
>
> **R2:** Please kindly note that we implement the latent space transform $z^*(\cdot)$ by updating $\mathbf{z}$ using I-FGSM rather than a neural network. Using an additional network to transform $\mathbf{z}$ does not offer any help to address the quality discontinuity issue (and indeed could be done within the generator), because the task of $z^*(\cdot)$ is to transform the continuous Gaussian distribution to $p_z^{op}$ (defined in Definition 3.3) which is supported on disconnected manifolds, thus $z^*(\cdot)$ must be a discontinuous mapping. However, neural network naturally leads to a continuous mapping. As an implicit iterative updating transform implemented for $z^*(\cdot)$ in our paper, it is capable of establishing a discontinuous mapping, and note the quantitative and qualitative results in Table 3, Figure 4 and Figure 9 have shown its significant effectiveness.
>
> **Q3: Ablation study to show the effectiveness of -z.**
>
> **R3:** Thanks for your suggestion. Indeed we have already conducted ablation studies in Table 4, 5 and 6. Please refer to Line 255-260 for the experimental setting: "-z" is the sampling improvement method; "-qua" and "-div" are the GAN training methods; "-qua+" and "-div+" are achieved by integrating "-qua" and "-z" and integrating "-div" and "-z". Taking Table 4 as an example, for each backbone, we report the results of both "-qua" and "-qua+". The comparison between the "-qua" column and the "bare" column shows the effectiveness of "-qua", while the comparison between the "-qua+" column and the "-qua" column shows the effectiveness of "-z".
>
> **Q4: It would be great to see the performance of AdvLatGAN on some larger-scale image dataset.**
>
> **R4:** Please refer to Line 316-329 and Table 6 for experimental results on large-scale datasets including ImageNet, CelebA and LSUN, where our methods achieve significant performance gain as always (-qua+ has achieved the best performance gain on FID in SNGAN LSUN-64 setting from 11.961 to 7.285). In rebuttal, we also supplement our new experiments on StyleGAN2-ada backbone on AFHQ and MetFaces datasets. Please refer to Appendix K.1.3 and Table 25 for the details.
>
> ---
> We hope this response could help address your concerns. We believe our work can contribute to the research in GAN with methodological novelty and strong empirical results. We would sincerely appreciate it if you could reconsider your rating and wish to receive your further feedback soon.

---

### Official Review · Reviewer_qXs3 · 2022-07-12

**Rating:** 5
**Confidence:** 4
**Soundness:** 2 fair
**Presentation:** 1 poor
**Contribution:** 2 fair

**Summary:**

This paper is about GAN training. The authors suggest that using a continuous latent distribution to model a discontinuous image distribution can lead to poor sample quality. The propose doing gradient updates directly on the latent representation to improve sample quality. They also do latent gradient updates to improve sample diversity. The authors show quantitative results on several natural image datasets, including ImageNet.

**Questions:**

Please could the authors clarify what they mean in lines 30-31 when they say that the generation quality is not a continuous function?

Line 51-52: “z* can be achieved by updating the latent z~p(z)” Can the authors please clarify that z* is a gradient update function on z? Or is it something else? Does z* have any parameters?

Should equation 2 be arg min?

Section 3.1 is confusing because it talks about both adversarial attacks and Generative Adversarial Networks which are different research areas that share the term “adversarial”. I would advise the authors to improve the clarify of this section.

How do equations (1) and (2) related to equations (3) and (4)?

Please clarify sentence on line 151-152: If all sub-manifolds M’s “are disconnected” how can the authors “require the splitting of Xr to the extent that each sub-manifold keeps connected.”

I suggest that the authors more clearly introduce the terms “latent space mining”, “adversarial sample mining” and how this is different to drawing samples from a generative model.

Proposition 3.2 is intuitive and makes sense.

Section 3.3: Are results on Table 1 computed across the whole test dataset? Results in Figure 6 do not suggest that there is an increase in sample diversity.

Results in Figure 5 are very nice and demonstrate the motivation (relating to sample quality) of the paper well. Please clarify in the caption which loss z* is being optimised for.

Line 243 “improved generation mapping G.” Does this method encourage the learning of better parameters in the generator, G? Or is it that you are able to draw better samples? Please clarify? If it is the former, how does your approach lead to improved learning of parameters for G?

Lin 254: Introduce acronyms IS and FID. For example, “Inception Score, SI [46], Fréchet Inception Distance, FID”.

Section 4.2. Which objective is being optimised for the results this section? Please reference the question in the text.

Results in Figure 9 are impressive. Are these cherry picked? Is it not possible to apply this to all steps in the interpolation from figure 3?

Line 301: Please clarify what is meant by “and does not work on unlabeled STL-10”.

Table 6: Please double check that all values show in bold are statically significantly better that the baselines. Where this is not the case either underline the others or indicate in some other clear way.

What is the application of this method? How does it affect the learned representations? Can they be used for more data efficient learning (for example in a classification task)?

**Limitations:**

Authors have considered a single limitation.
It would also be helpful to note how their approach affects the representation quality and the computation cost of optimising z*.

**Strengths And Weaknesses:**

Originality:
The authors draw our attention to an interesting problem, where by a continuous the latent distribution is used to model a discontinuous data distribution and propose using gradient updates directly on the latent representations to remedy this. The also do gradient updates on the latent space to improve variation. In both cases the authors use existing losses and optimise w.r.t. the latent representation.

Quality:
Figure 1 is helpful in motivating the problem that the authors are trying to solve.
Section 4.1: It is great that the authors consider many different variations of their model.
Impressive results in Figure 9 show how this approach can be used to "fix" poor samples. However, it would be better if we could see all the steps in the interpolation still and the cats ear is still partially missing.
Figure 4 is also great to show how the latent updates improve the image quality.
Results in Figure 8 further motivate the model.
Table 6 shows results on a large number of datasets. Its is good to see that the results hold on ImageNet.

While the authors have some good results the experimental setup is not well described, it is not always clear which objectives are being optimised.

There is no explanation of how a model with "in-training latent sampling transform" (AdvLatGAN-qua) performs well without "post-training latent sampling transform"? Intuitively it should be worse?

Addressing sample diversity is an after-thought and does not appear to be directly related to the main motivation of the paper.

Clarity:
This paper was very poorly written and very hard to follow. I have suggestions for improvements in the box below.

Significance:
The proposed work can be used to improve the quality of samples and diversity at the cost of doing gradient updates on the latent representations during sampling (although the authors do show results only doing updates durning training). This may be a high computation cost. It is also not clear what the application of this approach is. Does it improve sample efficiency during training on another task? How does the approach affect the learned representations?

It is also not clear to me why the problem of modeling a discontinuous data distribution is not solved by category conditional  generative models?

Strengths:
(1) The authors have very strong qualitative and quantitative results.

Weaknesses:
(1) The paper is very poorly written and very hard to follow.
(2) It is not clear what the application of this model is.

---

> ### Author Response · Authors · 2022-08-02
> **Response to Reviewer qXs3 (3/3)**
>
> **Q17: Line 308: Please clarify what is meant by “does not work on unlabeled STL-10”.**
>
> **R17:** Sorry for the ambiguity. ACGAN requires label input to generate. However, the experiment is based on the unlabeled part of STL-10 as it contains more images [1]. Thus ACGAN does not work on unlabeled STL-10, and we do not include it as one of the backbones in the STL-10 experiment. We have rephrased the statement as "We do not include ACGAN in the unlabeled STL-10 setting because it requires labels".
>
> [1] An Analysis of Single Layer Networks in Unsupervised Feature Learning AISTATS, 2011.
>
> **Q18: How does the approach affect the learned representations?**
>
> **R18:** For our task i.e. image generation, the generator learns how to map Gaussian noise to natural images. It does not involve learning representations (in common sense) of images. But if "representations" more generally refer to the outputs of the model i.e. generated results, then qualitative and quantitative results have shown our promising generation performance gain. This is achieved by our efficient latent space transform and a more powerful generative mapping trained by our algorithms (i.e. twofold efforts on $z^*$ and $G$), as claimed in the contributions and logical overview in our general response.
>
> We are open to further discussion on your question provided a more specific meaning of representation.
>
>
> **Q19: Other suggestions for improvements.**
>
> **R19:** Thanks for the nice suggestions. We list the improvements below.
>
> Equation 2: we have replaced "min" with "argmin".
>
> Figure 5: The optimization is guided by Eq. 5. We have added it in the caption.
>
> Line 261: Acronyms IS and FID have been introduced.
>
> Section 4.2 and 4.3: Optimised objectives or specific algorithms related to the sections have been additionally noted.
>
> ---
> We hope this response could help address your concerns. We believe our work can contribute to the research in GAN with methodological novelty and strong empirical results. We would sincerely appreciate it if you could reconsider your rating and wish to receive your further feedback soon.

---

> > ### Comment · Reviewer_qXs3 · 2022-08-08
> > **The role of equations (1) and (2) is still not clear.**
> >
> > Dear Authors,
> >
> > I appreciate the effort you have made to update your paper and improve the clarity. It is still not clear to me why the authors introduce equations (1) and (2) in the framework of an "adversarial attack"? Which paper is being referred to on line 139? It is also not clear what role these equations (1 & 2) play in the rest of the paper since they are not referenced in the rest of the paper.

---

> > > ### Author Response · Authors · 2022-08-08
> > > **Further response by authors**
> > >
> > > Thank you for the time and valuable comment. We have added the citations in Line 138-140. Below we explain the role of Eq. 1 and Eq. 2 in our paper.
> > >
> > > Eq. 1 and Eq. 2 generally express the basic idea of adversarial attack and adversarial training (**Eq. 1**: how to obtain adversarial samples through a constrained optimization; **Eq. 2**: adversarial training is achieved by using adversarial samples in the optimization of model training). Thus, Eq. 1 is largely related to AdvLatGAN-z, which introduces adversarial sample mining in latent space. And Eq. 2 is related to AdvLatGAN-qua and AdvLatGAN-div, which use transformed latent samples in the optimization of GAN. Please refer to Appendix E for more detailed illustration of the relation between our method and adversarial techniques.
> > >
> > > **Specifically**, **Eq. 1** is the general guidance of Eq. 5 and Eq. 8, where we replace the classification loss in Eq. 1 with the generator's loss and the mode seeking loss (i.e. Eq. 3). **Eq. 2** is the general guidance to AdvLatGAN-qua (Section 3.2.2 Line 202-214) and AdvLatGAN-div (Section 3.3 Line 219-221, 240-247), where we use the mined samples of Eq. 5 and Eq. 8 in GAN training. We have added the references to Eq. 1 and Eq. 2 in the corresponding sentences (Line 173, 203, 228, 245).
> > >
> > > We hope this response help to address your concern. If you have any further questions, please let us know.

---

> > > > ### Comment · Reviewer_qXs3 · 2022-08-09
> > > > **Thanks.**
> > > >
> > > > I have increased my score to 5.

---

> ### Author Response · Authors · 2022-08-02
> **Response to Reviewer qXs3 (2/3)**
>
> **Q9: Section 3.1 is confusing because it talks about both adversarial attacks and GAN which are different research areas that share the term “adversarial”.**
>
> **R9:** Thanks for noting. We have rephrased the statements to avoid confusion and further pointed out the roles of the adversarial attack/defense techniques in our methodology in Section 3.1. We use "bi-level optimization" to describe the adversarial process of GAN in our revision.
>
> **Q10: How do equations (1) and (2) related to equations (3) and (4)?**
>
> **R10:** Eq. 1 and 2 belong to the "Adversarial samples and adversarial training" part, while Eq. 3 and 4 belong to the "Mode coverage by regularizing distance of generated samples" part. They are two individual preliminaries and yet have no clear relations. Note Section 3.3 presents the logic of introducing adversarial techniques into MSGAN, obtaining the AdvLatGAN-div algorithm.
>
> **Q11: Line 158-159: If all sub-manifolds M’s “are disconnected” how can the authors “require the splitting of Xr to the extent that each sub-manifold keeps connected.”**
>
> **R11:** Here we mean sub-manifolds are disconnected from each other (about the relation between sub-manifolds), but we require that each noted sub-manifold is itself a connected set (about one single sub-manifold). Sorry to cause confusion. We have rephrased the statement.
>
> **Q12: I suggest that the authors more clearly introduce the terms “latent space mining”, “adversarial sample mining” and how this is different to drawing samples from a generative model.**
>
> **R12:** Thanks for the suggestion. We have added in Section 2 the explicit explanation of "latent space mining" and "adversarial sample mining" and their logic relation in this paper. "Latent space mining" is to modify **latent space samples** to achieve specific targets e.g. latent space sampling improvement for better generation, while "adversarial sample mining" methods investigate how to manipulate samples **by adding indistinguishable perturbations to cause huge network performance variance**. In this paper, "adversarial sample mining" is the tool we introduce in latent space to conduct sample mining/shifting.
>
> **Q13: Section 3.3: Are results on Table 1 computed across the whole test dataset?**
>
> **R13:** The results are computed over 5k generated images. This detail has been added to the paper.
>
> **Q14: Results in Figure 6 do not suggest that there is an increase in sample diversity.**
>
> **R14:** Figure 6 tends to show Eq. 8's effectiveness in obtaining pairs that tend to collapse (i.e. $\mathbf{z}$ distant in latent space and $G(\mathbf{z})$ close in pixel space) as we aim to regularize $G$ more purposefully through the aforementioned hard samples (sample pairs that tend to collapse) in training to improve $G$ (Line 219-222, 240-244). On the contrary, Eq. 8-inv tends to obtain pairs with much different generation results. We realize using "diverse" to describe this pair is imprecise, so we rephrase Line 239 and describe it as "leads to more different generation".
>
> **Q15: Does this method encourage the learning of better parameters in the generator, G? Or is it that you are able to draw better samples?**
>
> **R15:** Sorry for causing the confusion. In this paper, we make both efforts on drawing better samples and encouraging the learning of better parameters in the generator. Please refer to our rephrased Line 255-260 for the summarized description of the proposed variants. The GAN training methods using in-training sampling transform (i.e. -qua and -div) are GAN training methods to train better $G$. Please also refer to our general response for our contributions and logical overview.
>
> **Q16: Results in Figure 9 are impressive. Are these cherry picked? Is it not possible to apply this to all steps in the interpolation from figure 3?**
>
> **R16:** The results are not cherry-picked and we will definitely opensource our algorithm in our final version. Note that Fig. 9 is coupled to Fig. 3, and we directly use the bad results in Fig. 3 (Column 1-3) for improvement in Fig. 9, to testify our approach. AdvLatGAN-z aims to avoid bad generations caused by quality discontinuity (as shown in Fig. 1), thus we conduct the experiments on **invalid generations** of Fig. 3 (marked by yellow boxes). Please kindly note that this is not a cherry-picking. We have supplemented Figure 15 in the appendix as supplementary results to Figure 9, where we take Row 2 in Figure 3 as an example and apply AdvLatGAN-z to all steps in the interpolation.

---

> ### Author Response · Authors · 2022-08-02
> **Response to Reviewer qXs3 (1/3)**
>
> Thanks for the time, thorough comments and nice suggestions. We are pleased with your recognition to our qualitative and quantitative results. Below we respond to your comments/questions point-by-point, reclaim our technical contributions and application, clarify unclear/confusing statements and have revised the paper accordingly.
>
> **Q1: The application of the model is not clear.**
>
> **R1:** The motivation of the paper is to improve GAN models in a relatively rarely studied perspective which is also orthogonal to existing techniques. The image generation task we work on has a wide range of real-world applications and our model significantly enhances the generative performance on this task, which we believe contributes to the community. Please refer to the general response for more illustration.
>
> **Q2: It is not always clear which objectives are being optimized.**
>
> **R2:** Sorry for the confusion. We have added method tags to the Method section (Section 3.2.1, 3.2.2, 3.3), which we hope could help better correspond the method variants to the implementations. In addition, we have noted the corresponding optimized objectives in Figure 5's caption, Section 4.1 and specific algorithms in Section 4.2. Algorithm 1 and Algorithm 2 may also help to understand AdvLatGAN-qua and AdvLatGAN-div GAN training methods' optimization.
>
> **Q3: There is no explanation of how a model with "in-training latent sampling transform" (AdvLatGAN-qua) performs well without "post-training latent sampling transform"?**
>
> **R3:** Please refer to Section 3.2.2 for the explanation. In brief, AdvLatGAN-qua tries to mine samples that benefit the optimization and use them to calculate the optimization loss in GAN training, thus it can help train a more powerful $G$. The in-training methods are improving generative mapping $G$ and the post-training method are improving the latent variable sampling. They are orthogonal to each other, so AdvLatGAN-qua does not rely on AdvLatGAN-z to be effective.
>
> **Q4: The methods may have a high computation cost.**
>
> **R4:** The in-training algorithms do incur additional computational costs, but as we select a very small number of updating steps (1 for AdvLatGAN-qua and 3 for AdvLatGAN-div) in training, the additional relative overhead is small (around 20%-30%) while the performance gain is still significant. Please refer to Appendix D for the overhead experiments. Compared to the GAN's bi-level optimization process, the sampling method AdvLatGAN-z costs little, and it also outperforms peer latent sampling improvement methods in terms of cost-effectiveness in Table 8.
>
> **Q5: Does it improve sample efficiency during training on another task? Can the methods be used for more data efficient learning (for example in a classification task)?**
>
> **R5:** Our model mainly focuses on the task of image generation. Like many other variants of GAN, we tend to lift the generation performance of GAN and obtain more realistic generation. We believe our methodology novelty and promising results contribute to the community. Same thoughts may be able to be adapted to other tasks, but it is currently not the concern of this paper. We remain it for future thoughts or follow-up works.
>
>
> **Q6: It is also not clear to me why the problem of modeling a discontinuous data distribution is not solved by category conditional generative models?**
>
> **R6:** The issue can hardly be addressed by conditional generation: 1) there is no prior knowledge about the number of disconnected manifolds in the dataset. Please kindly note that there can also be many manifolds in a single category (Fig 3 indicates that quality discontinuity still exists in one category); 2) splitting the dataset and labeling the data according to the manifolds are intractable. Clearly our method offers a better solution.
>
> ---
> Below we respond to the questions/suggestions about presentation details.
>
> **Q7: Please could the authors clarify what they mean in lines 30-31 when they say that the generation quality is not a continuous function?**
>
> **R7:** The generation quality is reflected by the matching degree to the natural image distribution in pixel space. As natural image distribution is supported on disconnected manifolds, the generation quality does not exhibit a continuous nature, e.g. two nearby images' quality can differ much (please see Figure 1 and Figure 3). We have rephrased it in Line 31-32 to fix the ambiguity.
>
> **Q8: Line 55: Can the authors please clarify that** $z^*$ **is a gradient update function on z? Does** $z^*$ **have any parameters?**
>
> **R8:** Indeed we have noted in this sentence (Line 56-57) that $z^*$ is an I-FGSM updating function on $\mathbf{z}$ i.e. an implicit function achieved by several I-FGSM updates. Please refer to Eq. 6 for the formulas of one single update. $z^*$ has no parameters as it does not involve any additional network.

---

### Author Response · Authors · 2022-08-02
**General Response by Authors**

Dear area chair and reviewers,

We appreciate the reviewers' time and valuable comments. Overall, the reviewers generally agree to acknowledge our idea as interesting, novel or highly motivated, along with the comprehensive experiments and promising results. The major concerns lie in the confusion caused by unclear presentation, unclear logic flow and missing references on related works. Besides, additional experiments and overhead discussion are asked. To pre-clarify some potential misunderstandings that may affect the interpretation of our work, we first restate our contributions and try to resolve some general concerns.

**(I) Task/Application/Contributions.**
The motivation of the paper is to improve GAN models in a relatively rarely studied perspective which is also orthogonal to existing techniques. Specifically, we investigate GAN by rethinking the role of the generator as a continuous function and introducing adversarial learning in latent space to improve the generation performance. We show **two-fold efforts on both the latent space transform** $z^*(\cdot)$ **and the generative mapping $G(\cdot)$** while most pre-existing GAN variants merely focus on improving $G(\cdot)$: **1)** we introduce adversarial sample mining to achieve effective $z^*(\cdot)$; **2)** referring to adversarial training, we introduce latent transform in GAN training and effectively obtain more powerful generative mapping $G(\cdot)$. This also provides methodology guidance for further research of GAN variants. Experiments show that AdvLatGAN achieves significant performance gain and generates more realistic images.

**(II) Logical overview of the paper.**

1. The discontinuity in generation data quality and the continuity of mapping function requires a disconnected latent space, thus an effective discontinuous transform in latent space is needed. (Line 26-39)
2. The training algorithm and the trained generative mapping itself in terms of quality and diversity performance are also critical to the generation results. (Line 40-47)
3. Considering (1)(2), to achieve more realistic generation, twofold efforts on both $z^*(\cdot)$ and $G(\cdot)$ are required. (Line 48-51)
4. Since we hope that small perturbations of $\mathbf{z}$ could achieve considerable positive quality variation, we introduce adversarial sample mining method to achieve discontinuous transform $z^*(\cdot)$ in latent space. (Line 51-59)
5. Referring to adversarial training, we introduce targeted latent transform into GAN training to benefit optimization (i.e. using transformed latent variables to calculate the loss during GAN training), achieving more powerful $G(\cdot)$. (Line 60-66)

**(III) The pipeline for the generation of AdvLatGAN-qua+ and -div+.**

[Gaussian samples $\mathbf{z}$] -- AdvLatGAN-z --> [$z^*(\mathbf{z})$: transformed latent samples] -- $G$ trained by AdvLatGAN-qua or -div --> [$G(z^*(\mathbf{z}))$: final generation]

Please see Figure 7 for a schematic diagram.

For other issues that require clarification, we provide detailed answers in the following point-by-point responses and also supplement new experimental results to further verify our contributions. We have revised the paper according to the comments, with revisions marked in blue. Note that line numbers in the responses are all based on the revised version.

---

### Author Response · Authors · 2022-08-06
**Looking Forward to Your Feedback**

Dear Reviewers, given that there might exist some potential misunderstandings during the first round of review, we are sincerely looking forward to your reply and we could provide more information if needed.

---

### Author Response · Authors · 2022-08-09
**Inquiry for post-rebuttal comments**

Dear reviews,

We would like to express our sincere gratitude again for your valuable comments and nice suggestions. During the rebuttal phase, we have tried our best to eliminate the confusion and improve our paper. Since the discussion period is approaching its end, we would be glad to hear from you about whether our responses have addressed your concerns. We are sincerely looking forward to your further reply.

---

### Meta-Review · Area_Chair_qgtm · 2022-08-27

**Recommendation:** Accept
**Confidence:** Certain

**Metareview:**

The paper under consideration proposes to improve sample quality in GANs by performing optimization-based adversarial mining on the latent space as a pre-processing step, using the fast gradient sign method of Goodfellow et al (2015). Optimization in the latent space is not a new area, and I was surprised to see e.g. neither Bojanowski et al, 2017's "Optimizing the Latent Space of Generative Networks" nor Azadi et al, 2018's "Discriminator rejection sampling" cited as related ideas.

Reviewers found the idea interesting and the experiments mostly convincing, the contributions clear. The motivation for this paper hinges upon a proof under certain assumptions that complex distributions require a latent prior with disjoint/disconnected support; some reviewers were unclear how this was connected to FGSM but this was cleared up in rebuttal. Several reviewers expressed concerns about the far-from-state-of-the-art GAN "backbone"/small scale of the experiments, however new experiments involving larger resolution datasets and a StyleGAN2 base model have quelled these concerns.

To the AC, the method is intriguing, appears well-motivated and well-validated, especially in light of new experiments on larger scale data. Leveraging an adversarial example discovery procedure for the purpose of improving the prior sampling distribution is a clever and non-obvious innovation, and as the authors note in their rebuttal, while other feed-forward generative networks may suffer from the same issues around being "too continuous", the training procedure used by GANs (wherein the discriminator defines a non-stationary objective function, and its gradients are used as the generator's learning signal) uniquely position them to exploit this trick of performing "surgery" on the latent prior. For all of their difficulties, GANs have a reputation for accomplishing quite a lot in terms of sample fidelity for a fixed model capacity, and ideas like the one presented herein may serve to further alleviate training challenges, as well as elucidate niches in which GANs remain useful (at a time when the preponderance of attention has shifted to diffusion models). I recommend acceptance.

**Award:**

No

---

### Decision · Program_Chairs · 2022-09-14

Accept